# How realistic are the wakes of scaled wind turbine models?

Chengyu Wang[1], Filippo Campagnolo[1], Helena Canet[1], Daniel J. Barreiro[1], and Carlo L. Bottasso[1]

[1]Wind Energy Institute, Technische Universität München, D-85748 Garching b. München, Germany

**Correspondence:** Carlo L. Bottasso (carlo.bottasso@tum.de)

**Abstract.**

The aim of this paper is to analyze to which extent wind tunnel experiments can represent the behavior of full-scale wind turbine wakes. The question is relevant because on the one hand scaled models are extensively used for wake and farm control studies, whereas on the other hand not all wake-relevant physical characteristics of a full-scale turbine can be exactly matched by a scaled model. In particular, a detailed scaling analysis reveals that the scaled model accurately represents the principal physical phenomena taking place in the outer shell of the near wake, whereas differences exist in its inner core. A large eddy simulation actuator line method is first validated with respect to wind tunnel measurements, and then used to perform a thorough comparison of the wake at the two scales. It is concluded that, notwithstanding the existence of some mismatched effects, the scaled wake is remarkably similar to the full-scale one, except in the immediate proximity of the rotor.

## 1 Introduction

The simulation of wind turbine wakes in wind tunnels has been gaining an increasing interest in recent years. In fact, since wakes represent a major form of coupling within a wind plant, understanding their behavior and accurately simulating their effects are today problems of central importance in wind energy science, with direct practical implications on design, operation and maintenance. Recent studies include the analysis of single and multiple interacting wakes (see, for example, the review in Bottasso and Campagnolo (2020) or, among others, Whale et al. (1996); Chamorro and Porté-Agel (2009, 2010); Bartl and Sætran (2016); Bastankhah and Porté-Agel (2016); Tian et al. (2018); Campagnolo et al. (2016); Bottasso et al. (2014a); Campagnolo et al. (2020); Wang et al. (2020c) and references therein).

Wind tunnel testing offers some unique advantages over full-scale field testing:

– The ambient conditions are repeatable and —at least to some extent— controllable.

– Detailed flow measurements are possible with a plethora of devices, from standard pressure and hot-wire probes, to PIV (Meinhart, 1999) and scanning lidars (van Dooren et al., 2017), whereas measurements of comparable accuracy and resolution are today hardly possible at full scale. Additionally, time flows faster in a scaled experiment than at full scale (Bottasso and Campagnolo, 2020; Canet et al., 2020; Campagnolo et al., 2020), which means that a large informational content can be accumulated over relatively short periods of time.

– Models can be designed ad hoc to achieve specific goals, and can be extensively instrumented (Bottasso and Campagnolo, 2020), while layouts and scenarios can be readily changed to explore different operating conditions of interest.

   – Costs are limited, even for highly sophisticated models, also because there are no energy production losses as it is often the case in the field; additionally, the costs of sophisticated wind tunnel facilities are typically amortized by their use for several different applications over long periods of time.

– Open datasets can be shared within the research community and collaborations are facilitated, since there are no —or fewer— constraints from intellectual property than when real wind turbine data is used.

Testing in the controlled and repeatable environment of the wind tunnel is today contributing to the understanding of the physical processes at play, generates valuable data for the validation and calibration of mathematical models, and offers opportunities for the verification of control technologies.

However, notwithstanding these and other unique advantages, a major question still hovers over the wind tunnel simulation of wakes: *how faithful are these wakes to the actual ones in the field?* In fact, in private conversations these authors have often been questioned on the actual usefulness of wind tunnel testing, based on a perceived lack of realism of these scaled experiments. Indeed, some skepticism is justified and completely understandable: simulation codes are being calibrated and validated with respect to wind tunnel measurements, and wind farm control techniques are being compared and evaluated in

wind tunnel experiments. Therefore, it is important to quantify the level of realism of wind tunnel simulated wakes, and to identify with better clarity what aspects faithfully represent the full-scale truth and what aspects do not.

A thorough and complete answer to this question is probably still out of reach today. In fact, detailed inflow and wake measurements of a full-scale turbine would be necessary, with a level of detail comparable to the ones achievable in the tunnel. Lidar technology is making great progress (Zhan, 2020), and might soon deliver suitable datasets. It should be a goal of the

scientific and industrial communities to completely open such future datasets to research, which would surely greatly favor the scientific advancement of the field. In the meanwhile, however, some partial answers to the question of wake realism can still be given. This is the main goal of the present paper.

This study considers the TUM G1 scaled wind turbine (Bottasso and Campagnolo, 2020), and a dataset obtained with this machine in the boundary layer wind tunnel of the Politecnico di Milano in Italy. A large eddy simulation (LES) actuator line

method (ALM) (Wang et al., 2019) is used to simulate the wind tunnel experiments, including the passive generation of a sheared turbulent inflow. The code has been validated with respect to the present and other similar measurements.

Following Bottasso and Campagnolo (2020) and Canet et al. (2020), dimensional analysis and wake physics are used here to review the main factors driving wake behavior. The same analysis also reveals which physical aspects of full-scale wakes cannot be matched at the reduced scale and with the considered experimental setup. A first analysis of scaling was performed

by Chamorro et al. (2016), considering the effects caused by the mismatch of the rotor-based Reynolds. Experimental results based on a miniature wind turbine showed that wake behaviour is unaffected by this parameter when it is larger than circa $10^5$. However, in reality the behavior of the blades and, as a consequence, of the wake is much more strongly affected by the chord-based Reynolds number, as initially discussed in Bottasso et al. (2014a). In fact, the much lower Reynolds regime of a

small-scale model blade compared to a full-scale machine implies very different aerodynamic characteristics of the airfoils,
which in turn drive a number of specific design choices of the scaled model (Bottasso and Campagnolo, 2020; Canet et al.,
2020). Notwithstanding the differences caused by the chord-based Reynolds number mismatch, it is relatively easy —as shown
more in detail later on— to match the main processes taking place in the outer shell of the near wake, as well as the ones that
govern its breakdown and the characteristics of the far wake. On the other hand, several mismatched effects do exist in the
central core of the near wake. Dimensional analysis also expresses the scaling relationships that allow the mapping of scaled
quantities into equivalent full-scale ones, and viceversa.

Based on the understanding provided by dimensional analysis and wake physics, full-scale turbines are designed in this
work to match some of the G1 scaled-model parameters. Various versions of these models are considered, ranging from a more
realistic full-scale turbine —with a larger number of mismatched effects— to less realistic ones that however match a larger
set of quantities of the scaled model.

The full-scale models are then simulated with the LES-ALM code, using the same exact numerical methods and algorithmic
parameters used for the scaled simulations. These wind turbine models are also exposed to the same identical ambient turbulent
inflow used for the scaled model. The underlying assumption is that, since the code was found to be in very good agreement
with measurements obtained in the scaled experiments, the same code based on the same numerical setup should deliver results
of similar accuracy even at full scale. This assumption cannot be formally proven at this stage, but it seems to be very reasonable
and it is probably the only possible approach that can be pursued in the absence of a detailed full-scale dataset.

Finally, the numerically simulated scaled and full-scale wakes are compared. The analysis considers wind-aligned and mis-
aligned conditions, typical of wake steering control applications, and various metrics, including wake shape, path, speed profile,
Reynolds shear stresses, power available and wind direction modification due to the curled wake in misaligned conditions. This
detailed comparison is used to quantify the degree of similarity among the different models and across the various metrics.
Since the models differ by known mismatched effects, this also helps pinpoint and explain any source of discrepancy.

The paper is organized according to the following plan. Section 2 uses dimensional analysis and wake physics to identify
the quantities that can be exactly matched between scaled and full-scale models, the ones that can only be partially matched,
the ones that are unmatched, and those that are neglected from the present analysis. Next, Section 3 describes the scaled
experimental wind turbine and its full-scale counterparts, which include various modifications to highlight the effects of specific
mismatches. Section 4 describes the numerical simulation model, including the generation of the turbulent inflow in the wind
tunnel. Results and detailed comparisons among the scaled and the full-scale models are reported in Section 5. Finally, Section 6
summarizes the main findings of this work.

## 2 Scaling

The matched, partially matched, unmatched and neglected physical effects of the scaled and full-scale models are reviewed
next. Quantities referred to the scaled model are indicated with the subscript $(\cdot)_M$, while quantities referred to the full-scale
physical system with the subscript $(\cdot)_P$. Scaling is defined by two parameters (Bottasso and Campagnolo, 2020; Canet et al.,

2020): the length scale factor $n_l = l_M/l_P$, where $l$ is a characteristic length (for example the rotor radius $R$), and the time compression ratio $n_t = t_M/t_P$, where $t$ is time. In the present case $n_l = 1/162.1$ and $n_t = 1/82.5$. A more complete treatment of scaling for wind turbine rotors is given in Bottasso and Campagnolo (2020) and Canet et al. (2020).

## 95 2.1 Matched quantities

- *Inflow*. The ambient flow is obtained by simulating the passive generation of turbulence in the wind tunnel, as explained in §4.2; the developed flow is sampled on a rectangular plane, which becomes the inflow of the scaled turbine simulations. For the full-scale turbine simulations, the sides of the rectangular inflow area are geometrically scaled by $n_l$, while time is scaled by $n_t$ and speed $V$ as $V_M/V_P = n_l/n_t$, resulting in a flow with exactly the same identical characteristics (e.g., shear, turbulence intensity, integral length scale, etc.) at the two scales.

- *Tip speed ratio (TSR)* $\lambda = \Omega R/V$, where $\Omega$ is the rotor speed. TSR determines not only the triangle of velocity at the blade sections, but also the pitch of the helical vortex filaments shed by the blade tips.

- *Non-dimensional circulation* $\Gamma(r)/(RV) = 1/2\,(c(r)/R)\,C_L(r)(W(r)/V)$, where $C_L$ is the lift coefficient, $c$ the local chord, $W$ the local flow speed relative to the blade section, and $r$ is the spanwise blade coordinate (Burton et al., 2011). Each blade sheds trailing vorticity that is proportional to the spatial (spanwise) gradient $\mathrm{d}\Gamma/\mathrm{d}r$. Therefore, matching the non-dimensional spanwise distribution of $\Gamma$ (and, hence, also its non-dimensional spanwise gradient) ensures that the two rotors shed the same trailing vorticity.

  The root of the G1 blade is located further away from the rotor axis than a typical full-scale machine, due to the space required for housing the pitch actuation system in the hub. The resulting effects caused on the wake were investigated by developing two different full-scale models: one with the exact same non-dimensional circulation of the G1, and one with more typical full-scale values, as discussed later.

- *Rotor-based Strouhal number*. The rotor-based Strouhal number $\mathrm{St} = fD/V$ is matched, where $f$ is a characteristic frequency and $D = 2R$ is the rotor diameter. This definition of the Strouhal number has been recently shown to characterize the enhanced wake recovery obtained by mixing, both in the case of dynamic induction control (Frederik et al., 2020a) and by cyclic pitch excitations (Frederik et al., 2020b).

## 2.2 Approximatively matched quantities

The following quantities or effects are very nearly, but not exactly, matched:

- *Thrust coefficient* $C_T = T/(1/2\rho AV^2)$, where $T$ is the thrust force, $\rho$ is air density and $A = \pi R^2$ the rotor swept area. The thrust characterizes to a large extent the speed deficit in the wake. In misaligned conditions, it is also the principal cause for the lateral deflection of the wake. The thrust coefficient is very nearly matched, whereas the power coefficient is not (as discussed later). In fact, the latter strongly depends on airfoil efficiency, which is affected by the chord-based

Reynolds number mismatch between the two models. On the other hand, drag has only a limited effect on thrust, which as a result is very similar in the models at the two scales.

– *Dynamic spanwise vortex shedding.* During transients, spanwise vorticity is shed that is proportional to the temporal gradient of the circulation. To match the spanwise vortex shedding of a rotor, the matching of $(1/RV)\mathrm{d}\Gamma/\mathrm{d}\tau$ should be ensured (Bottasso and Campagnolo, 2020; Canet et al., 2020), where $\tau$ is a non-dimensional time (for example, $\tau = \Omega_r t$, $\Omega_r$ being a reference rotor speed), equal for both the full and scaled models.

Rewriting the non-dimensional circulation as

$$\frac{\Gamma}{RV} = \frac{1}{2}\frac{c}{R}C_{L_\alpha}\frac{W}{V}\left(\frac{U_P U_T}{W^2} - \theta\right), \tag{1}$$

$C_{L_\alpha}$ being the lift curve slope, the dynamic spanwise vortex shedding condition implies the matching of the non-dimensional time rates of change of the sectional tangential and perpendicular flow components $U_P$ and $U_T$, with $W^2 = U_P^2 + U_T^2$, and of the pitch angle $\theta$. The flow speed component tangential to the rotor disk is $U_T = \Omega r + u_T$, where $u_T$ contains terms due to wake swirl and yaw misalignment. The flow speed component perpendicular to the rotor disk is $U_P = (1-a)V + u_P$, where $a$ is the axial induction factor, and $u_P$ the contribution due to yaw misalignment and vertical shear. A correct similitude of dynamic vortex shedding is ensured if the non-dimensional time derivatives $\lambda'$, $a'$, $u_P'$, $u_T'$ and $\theta'$ are matched, where $(\cdot)' = \mathrm{d} \cdot /\mathrm{d}\tau$.

Matching of $\lambda'$ is ensured here by the fact that the two rotors operate at the same TSR in the same inflow; additionally, the simulations were conducted by prescribing the rotor rotation (i.e. without a controller in the loop), so that $\Omega' = 0$. The term $a'$ accounts for dynamic changes in the induction, which are due to the speed of actuation (of torque and blade pitch) and by the intrinsic dynamics of the wake. The speed of actuation is not relevant in this case, due to the absence of a pitch-torque controller. The intrinsic dynamics of the wake, as modelled by a first order differential equation (Pitt and Peters, 1981), is also automatically matched thanks to the matching of the design TSR (Bottasso and Campagnolo, 2020; Canet et al., 2020). Finally, $u_P'$ and $u_T'$ are matched because the inflow is the same, with the exception of the contribution of wake swirl, which is not exactly the same because of the different torque coefficient, as noted below.

– *Inflow size.* The cross section of the wind tunnel has a limited size, resulting in the blockage phenomenon, i.e. in an acceleration of the flow between the object being tested and the sides (lateral walls and ceiling) of the tunnel (Chen and Liou, 2011). Although this problem is not strictly related to the scaling laws discussed here, it is still an effect that needs to be accounted for, especially if the ratio of the frontal area of the tested objected and the cross sectional area of the tunnel is not negligible. Simulations in domains of increasingly larger cross sections were conducted to quantify the blockage affecting the experimental setup considered here.

– *Integral length scales (ILS).* Relative to the size of the TUM G1 turbines, the wind tunnel used in this research (located at Politecnico di Milano, Italy) generates a full-scale ILS of approximately 142 m at hub height, which is respectively about 16% and 58% smaller that the lengths specified by Ed. 2 (IEC 61400-1, 1999) and Ed. 3 (IEC 61400-1, 2005) of

the IEC 61400-1 international standards. To understand the effects of this mismatch on wake behavior, simulations were conducted in turbulent inflows differing only in their integral scales.

## 2.3 Unmatched quantities

The following quantities cannot be matched based on the current experimental setup and scaling choices:

– *Chord-based Reynolds number* $\mathrm{Re} = \rho W c/\mu$, where $\mu$ is the fluid viscosity. The Reynolds number mismatch can be computed as $\mathrm{Re}_M/\mathrm{Re}_P = n_l^2/n_t$, which is equal to $1/318.5$ in the present case. This implies that the blades of the G1 model operate in a very different regime than the ones of the full-scale blade (Lissaman, 1983). To mitigate these effects, the G1 blade has a larger chord than the full-scale one, and uses ad hoc low-camber airfoils specifically conceived for low-Reynolds-number flows (Bottasso and Campagnolo, 2020; Selig, 2003). Additionally, note that the scaling relationship of the rotor speed is $\Omega_M/\Omega_P = 1/n_t$. Therefore, by increasing the rotor speed of the model $\Omega_M$ (which has the effect of accelerating time by reducing the ratio $n_t$), one can lower the Reynolds mismatch (Bottasso and Campagnolo, 2020).

– *Power coefficient* $C_P = P/(1/2\rho A V^3)$, where $P$ is the aerodynamic power. The power coefficient of the scaled model is lower than the one of the full-scale machine, because of the smaller efficiency of the airfoils at low-Reynolds regimes. Since the torque coefficient is $C_Q = C_P/\lambda$, then also $C_Q$ is unmatched and smaller for the scaled model than for the full-scale one, resulting in reduced wake swirling (Burton et al., 2011).

– *Tower and nacelle vortex shedding*. Bluff bodies periodically release vortices in their wakes (Karman, 1911), at a characteristic frequency proportional to the Strohual number. The tower-based Strouhal number $\mathrm{St} = fd/V$ is matched when the tower diameter $d$ is geometrically scaled. However, as noted later, the diameter of the G1 tower is larger than the one of the full-scale machine, so that frequency and size of the shed vortices are accordingly affected. An even larger mismatch applies to the nacelle, because of power density and miniaturization constraints.

– *Stall delay due to rotational augmentation* (Dowler and Schmitz, 2015). Matching these effects requires the matching of the blade chord and twist distributions, of the non-dimensional circulation, and of the Rossby number $\mathrm{Ro} = \Omega r/(2W)$ (Bottasso and Campagnolo, 2020). While the latter two quantities are indeed matched, the former two are not, in order to mitigate the chord-based Reynolds number mismatch. To quantify the effects of rotational augmentation on wake behavior, two versions of the full-scale turbine were developed, as explained later on.

– *Chord-based Mach number* $\mathrm{Ma} = W/s$, where $s$ is the speed of sound. Although this flow parameter is not matched, compressibility effects are irrelevant for the full and scaled models considered here, as for virtually all present-day wind turbines.

– *Boundary layer stability and wind veer due to the Coriolis force*. The wind tunnel used in the present research can only general neutrally stable boundary layers. Although atmospheric stability has a profound effect on wakes (Abkara and Porté-Agel, 2015), this problem has already been studied elsewhere, and it is considered to be out of scope for the present

investigation. Similarly, Coriolis effects on the inflow and wake behavior are not represented in a wind tunnel, although they are known to have non-negligible effects on capture, loading and also on wake path (van der Laan and Sørensen, 2007).

## 2.4 Neglected quantities

The following effects could be matched with a different experimental setup and scaling choices, but were neglected in the present work:

- *All gravo-aeroelastic effects*. Since the blades of the G1 turbine are not aeroelastically scaled (and are very stiff), also the full-scale model was simulated without accounting for flexibility. Aeroelasticity could have some effects on near-wake behavior for very flexible rotors, but would probably have only a negligible role on the characteristics of the far wake. Therefore, aeroelastic effects were excluded from the scope of the present investigation.

- *Unsteady airfoil aerodynamics*, including linear unsteady corrections (for example, according to Theodorsen's theory (Bisplinghoff and Ashley, 2002)), and dynamic stall. It was verified that the mildly misaligned operating conditions analyzed here would not have triggered dynamic stall, except than in a few instances, similarly to what was found in Shipley (1995). Here again, these effects would hardly have any visible effects on far-wake behavior.

## 2.5 Remarks

Wake stability analysis shows that the vortical structures released by the blade tips and root interact in the near wake (Okulov and Sørensen, 2007).

In the *outer shell* of the near wake, the mutual interaction of the tip vortices —triggered by turbulent fluctuations— lead to vortex pairing, leapfrogging, and eventually to the breakdown of the coherent wake structures (Sørensen, 2011). The scaled and full-scale rotors are exposed to the same inflow (including the same ambient turbulent fluctuations), the tip vortices have the same geometry (due to a matched design TSR) and strength (due to a matched non-dimensional circulation), and the speed deficit is also essentially the same (because of the very nearly matched thrust coefficient). Hence, it is reasonable to assume a nearly identical near-wake behavior of the external wake shell, given that all main processes are matched between scaled and full-scale models (with the exception of the effects that the unmatched tower may have).

The situation is different in the near-wake *inner core*. Here the root vortices combine with the effects caused by the presence of the nacelle and tower. In particular, the nacelle has a much larger relative frontal area, creating a different blockage (radial redirection), nacelle wake and vortex shedding. Additionally, in the 20% inboard portion of the blade, both the circulation and rotational augmentation effects are unmatched. Finally, the mismatch of power induces a mismatch of torque that reduces wake swirl; as shown by blade element momentum (BEM) theory, swirl is mostly concentrated in the inner core of the wake, and decays rapidly with radial position (Burton et al., 2011). Hence, the near-wake inner core is expected to behave differently in the scaled and full-scale models. However, some of the results reported here, in addition to evidence from other sources (Wu and Porté-Agel, 2011), indicate that the inner core near wake has only a modest effect on far-wake behavior. For example, it

is common practice to simulate far-wake behavior with LES codes without even representing the turbine nacelle and tower (Martínez-Tossas et al., 2015).

As a consequence, thanks to the employed scaling and matching criteria, the far-wake behavior is expected to be extremely
similar between the wind-tunnel-generated wake and the full-scale one. The results section will more precisely support this claim.

## 3 Wind turbine models

### 3.1 The TUM G1 scaled wind turbine

The TUM G1 is a three-bladed clockwise-rotating (looking downstream) wind turbine, with a rotor diameter D of 1.1 m, a hub
height H of 0.825 m, and rated rotor and wind speeds of 850 rpm and 5.75 ms$^{-1}$, respectively. The G1 was designed based on
the following requirements (Bottasso and Campagnolo, 2020):

- – A realistic energy conversion process and wake behavior;

- – A sizing of the model obtained as a compromise between Reynolds mismatch, miniaturization constraints, limited wind
  tunnel blockage, and ability to simulate multiple wake interactions within the size of the test chamber;

- – Active individual pitch, torque, and yaw control in order to test modern control strategies at the turbine and farm levels;

- – A comprehensive on-board sensorization.

The turbine has been used for several research projects and numerous wind tunnel test campaigns (Campagnolo et al., 2016,
2020). The main features of the G1 rotor and nacelle are shown in Fig. 1a.

A brushless motor equipped with a precision gearhead and a tachometer is installed in the rear part of the nacelle and
generates the resisting torque, which is in turn measured by a torque sensor located behind the two shaft bearings. An optical
encoder, located between the slip ring and the rear shaft bearing, measures the rotor azimuth, while two custom-made load
cells measure the bending moments at the foot of the tower and on the shaft in front of the aft bearing. Thrust is estimated from
the tower-base fore-aft bending moment, correcting for the drag of the tower and rotor-nacelle assembly.

Each wind turbine model is controlled by its own dedicated real-time modular Bachmann M1 system, implementing super-
visory control functions, pitch-torque-yaw control algorithms, and all necessary safety, calibration and data logging functions.
Measurements from the sensors and commands to the actuators are transmitted via analogue and digital communication. The
Bachmann M1 system is capable of acquiring data with a sample rate of 2.5 kHz, which is used for aerodynamic torque, shaft
bending moments and rotor azimuth position. All other measurements on the turbine are acquired with a sample rate of 250 Hz.

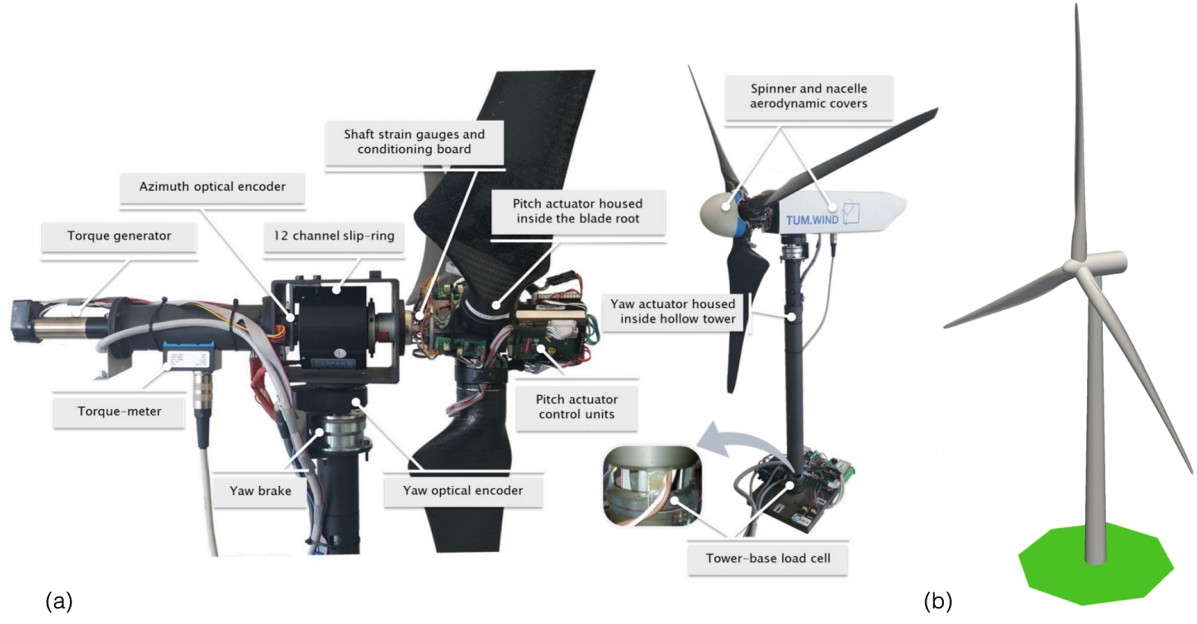

**Figure 1. (a)** The TUM G1 turbine (Campagnolo et al., 2016). **(b)** The full-scale DTU 10 MW turbine (from Bak et al. (2013)).

## 3.2 Full-scale wind turbine

A full-scale wind turbine was designed through a backward-engineering approach to match the characteristics of the G1 scaled machine. The DTU 10 MW wind turbine (Bak et al., 2013), shown in Fig. 1b, was used as a starting design for this purpose. This turbine has a rotor diameter of 178 m and a hub height of 119 m, and the modified version used here is termed G178.

The ratio of the rotor diameter D of the G1 and DTU turbines was used to define the geometric scaling factor $n_l$. The hub height H of the full-scale machine was slightly adjusted to match the ratio D/H of the G1 turbine.

The shape of nacelle and tower were kept the same as the DTU reference, creating a mismatch with the G1 turbine. In fact, the scaled model —due to miniaturization constraints— has a frontal area of the nacelle that is 2.6 times larger than the one of the scaled DTU turbine; similarly, the tower diameter of the G1 turbine is 49% larger than the scaled one of the DTU machine. This creates a mismatch in the drag of the nacelle and tower, in their local blockage and vortex shedding.

The aerodynamic design of the rotor of the DTU turbine was modified, in order to match the characteristics of the G1 in
terms of design TSR and non-dimensional circulation distribution (and, as a consequence, to match also the thrust). Three versions of the rotor were realized. The standard G178 uses the same airfoils of the DTU turbine over the entire blade span, while chord and twist distributions were modified to satisfy the matching criteria. As the root of the G1 blade is located further away from the rotor axis than in the case of the G178, the non-dimensional circulation is matched only between 20% and 100% of blade span. To account for the effects of rotational augmentation, the inboard airfoils were corrected for delayed stall
according to the model of Snel (1994).

A second rotor was designed to investigate the effects of the mismatched non-dimensional circulation on wake behavior. To this end, the twist angle close to the root was modified to decrease the lift inboard and match the non-dimensional circulation of the G1 turbine even in this part of the blade; all the other parameters of the model were kept the same as in the G178 turbine. This second turbine is termed G178-MC, where MC stands for 'matched circulation'.

A third version of the rotor was obtained by eliminating from the G178 the rotational augmentation model, to investigate its effects. The resulting rotor is termed in the following G178-nRA, where nRA stands for 'no rotational augmentation'.

The blades of the reference turbine are equipped with the four airfoils FFA-W3-241, FFA-W3-301, FFA-W3-360, and FFA-W3-480 (Fuglsang et al., 1998), respectively from tip to root. For the operating conditions analyzed in this paper, the chord-based Reynolds of the G1 varies along the blade span within the range 60,000-85,000. Airfoils operating at a Reynolds number below 100,000 experience significant parasitic drag due to the formation of a laminar separation bubble (Winslow et al., 2018), which affects their maximum lift coefficient and lift-to-drag ratio. To limit these effects, the low-Reynolds airfoil RG14 (Lyon and Selig, 1996) is used throughout the whole span of the G1. Trips can be employed for triggering the boundary layer transition and eliminate or reduce the laminar bubble (Selig and McGranahan, 2004). However, tripping is not used on the G1 blades, because it is not effective on these low-camber airfoils (Campagnolo, 2013).

The efficiency $E = C_L/C_D$ vs. non-dimensional span $r/R$ of the reference and scaled blade is shown in Fig. 2 at rated conditions. The airfoil efficiency for the scaled rotor is almost half the one of the full-scale machine in the outer span of the blade; since most of the power is indeed extracted in this region, the reduced efficiency results in a lower power coefficient for the scaled model. The FFA-series airfoil characteristics were obtained with ANSYS Fluent (ANSYS, Inc., 2019), while the RG14 ones by correcting the baseline values of Lyon and Selig (1996) with rotor power and thrust measurements through the tuning approach of Wang et al. (2020a).

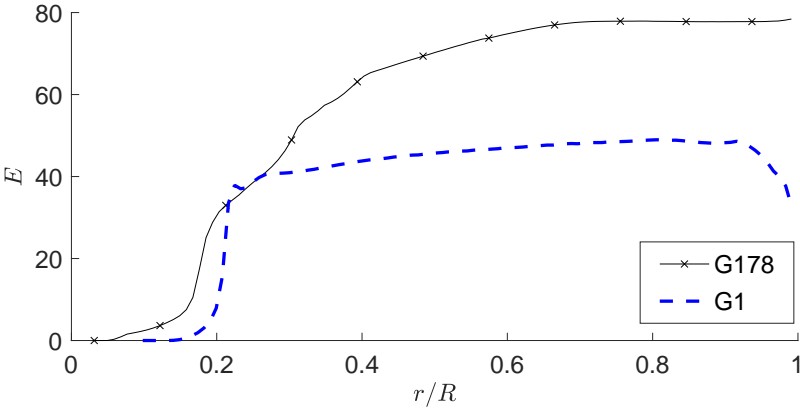

**Figure 2.** Efficiency $E$ along the blade span $r/R$ for the G178 and G1 turbines at rated TSR.

Distributions of the twist, chord, lift coefficient and non-dimensional circulation of the G1 and of the full-scale rotors are shown in Fig. 3. Chord distributions are normalized by their respective arithmetic mean $c_0$ over the span. Lift coefficient and

circulation are evaluated at rated conditions using the BEM method implemented in the code FAST 8 (Jonkman and Jonkman, 2018). The lift coefficient of the G1 is significantly smaller than the one of the full-scale turbines, which is a result of the its higher rotor solidity. The lower lift is however compensated by a larger chord and different twist distributions, resulting in a matched non-dimensional circulation from 20% span to the blade tip for the G178 turbine. For the G178-MC model, the non-dimensional circulation is matched over the whole blade span. The difference in lift and circulation between G178 and G178-nRA is due to rotational augmentation.

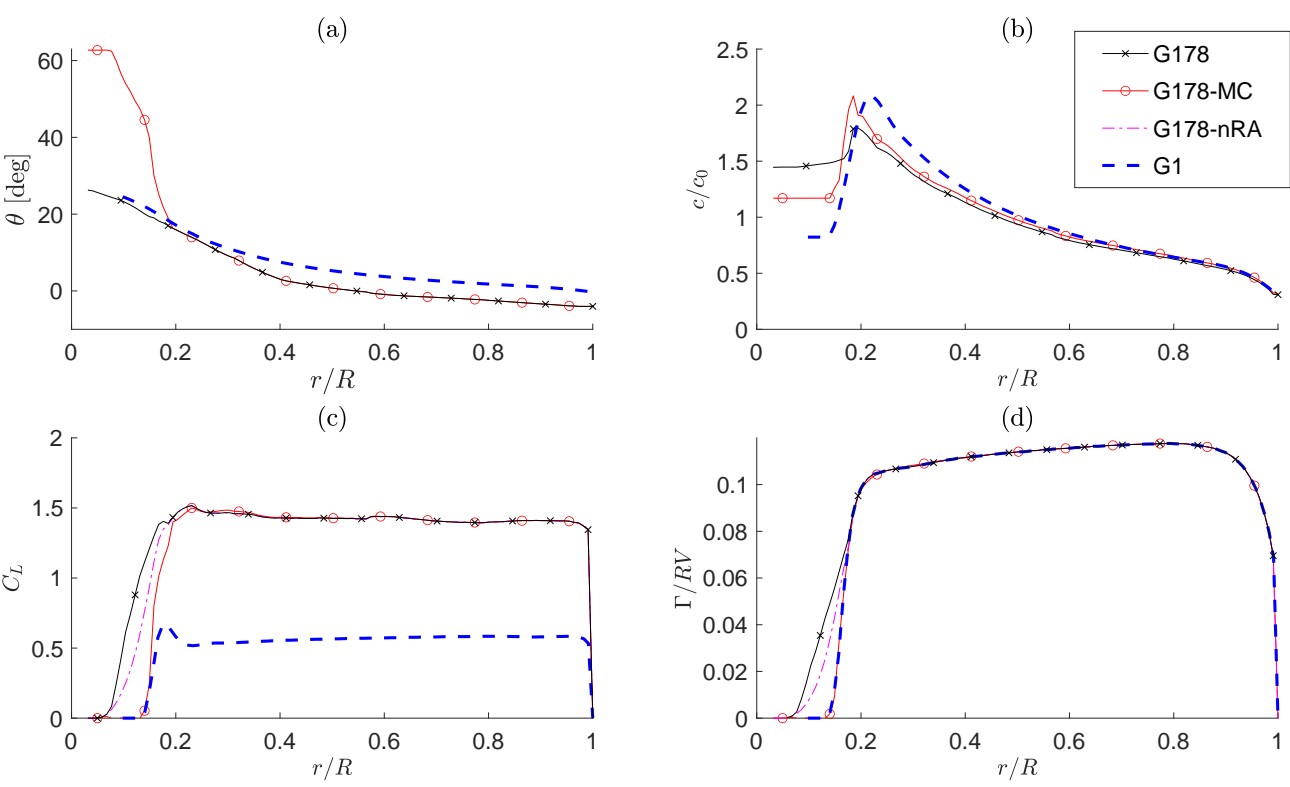

**Figure 3.** Distributions of twist $\theta$ **(a)**, non-dimensional chord $c/c_0$ **(b)**, lift coefficient $C_L$ **(c)**, and non-dimensional circulation $\Gamma/RV$ **(d)**, for the G1 and for the G178, G178-MC, and G178-nRA full-scale turbines.

# 4 Simulation model

## 4.1 LES-ALM CFD code

Numerical results were obtained with a TUM-modified version of SOWFA (Fleming et al., 2014), more completely described in Wang et al. (2018, 2019). The code has been used extensively to numerically replicate wind tunnel tests conducted with G1

turbines, achieving an excellent correlation with the experimental measurements in a wide range of conditions, including full and partial wake overlaps, wake deflection, static and dynamic induction control, and individual pitch control (for example, see Wang et al. (2019, 2020b, c)).

The finite volume LES solver is based on the standard Boussinesq PISO (Pressure Implicit with Splitting of Operator) incompressible formulation, and is implemented in OpenFOAM (Jasak, 2009). Spatial differencing is based on the Gamma method (Jasak et al., 1999), where a higher level of upwinding is used in the near wake region to enhance stability. Time marching is based on the backward Euler scheme. The pressure equation is solved by the conjugate gradient method, preconditioned by a geometric-algebraic multi-grid, while a bi-conjugate gradient is used for the resolved velocity field, dissipation rate and turbulence kinetic energy, using the diagonal incomplete-LU factorization as preconditioner. The turbulence model is based on Smagorinsky (1963), where the Smagorinsky constant is equal to 0.16.

An actuator-line method (ALM) (Troldborg et al., 2007) is used to represent the effects of the blades, according to the velocity sampling approach of Churchfield et al. (2017). The implementation of the actuator lines is obtained by coupling the CFD solver with the aeroservoelastic simulator FAST 8 (Jonkman and Jonkman, 2018). For improved accuracy, the airfoil polars of the G1 are tuned based on experimental operational data (Bottasso et al., 2014b; Wang et al., 2020a). The rotor speed is set to a constant value to precisely match the desired TSR (Wang et al., 2018). Finally, the immersed boundary (IB) formulation method (Mittal and Iaccarino, 2005; Jasak and Rigler, 2014) is employed to model the effects of the turbine nacelle and tower.

Details on the mesh and other algorithmic settings are described in the following sections.

## 4.2 Turbulent inflow

Experiments with the G1 turbine took place in the large boundary layer test section of the wind tunnel at the Politecnico di Milano, where a turbulent flow is generated passively by the use of spires. Without the spires, the flow at the inlet has a turbulence intensity (TI) of about 1-2% and a small horizontal variability caused by the presence of 14 fans and internal transects upstream of the chamber. The non-uniform blockage caused by the spires decelerates the flow close to the wind tunnel floor, generating an initial vertical shear; furthermore, large vortical structures develop around the edges of the spires, which then break down as the flow evolves moving downstream.

Two setups are considered, with two different TI levels. To mimic a typical medium-turbulence offshore condition, 14 type-B spires were placed side by side 1 m from each other, 1 m downstream of the test chamber inlet. A type-B spire consists of an equilateral trapezoid and a supporting board. The height of the trapezoid is 2.0 m, while the widths of the bottom and top edges are 0.26 m and 0.1 m, respectively. The developed turbulent flow where the turbine is located (19.1 m downstream of the inlet) has a vertical shear with a power coefficient equal to 0.12, a small horizontal shear, and hub-height speed and TI of 5.75 ms$^{-1}$ and 5%, respectively. A second higher-turbulence inflow was generated using 9 triangular spires with a height of 2.5 m and a base of 0.8 m, placed at a distance of 1.55 m from each other. In addition, 24 rows of $0.23 \times 0.23 \times 0.1$ m bricks were placed on the ground, with 12 bricks in odd rows and 13 bricks in even ones, resulting in a staggered brick distribution. This second

configuration resulted in a vertical shear with a power coefficient equal to 0.19, a small horizontal shear, and hub-height speed and TI of 5.75 ms$^{-1}$ and 14%, respectively.

The simulations were conducted in two phases: first, developed turbulent flows were obtained by simulating the interaction of the chamber inlet wind with the spires and bricks; next, the results of these precursor simulations were sampled on a plane 3.59 D upstream of the rotor disk, and used as inlet for the simulations of the turbine and its wake. For the turbulence-generating precursor simulations, the mesh was obtained with ANSYS-ICEM, which resulted in a structured body-conforming grid around the spires (Wang et al., 2019), entirely consisting of hexahedral elements. The bricks placed on the floor for the higher turbulence case were modelled by the IB method. All simulations included the floor, side walls and the ceiling of the tunnel. Boundary layers on these surfaces were modeled by wall functions with an average $y^+$ value of 50, achieved with local mesh refinement. The chamber cross section has a width of 13.84 m and a height of 3.84 m, resulting in some vertical blockage, whose effects were quantified by running various simulations for increasing values of the chamber height, as reported later.

The grid for the wind turbine simulations uses three zones of increasing density, as shown in Fig. 4, the smallest cells having a size of 0.015 m (i.e., $1.4 \cdot 10^{-2}$ D). The ALM discretization used 108 points over the blade span, i.e. a spacing equal to $4.7 \cdot 10^{-3}$ D. The simulations were run for 360 rotor revolutions, which were enough for reaching a turbulent steady-state regime.

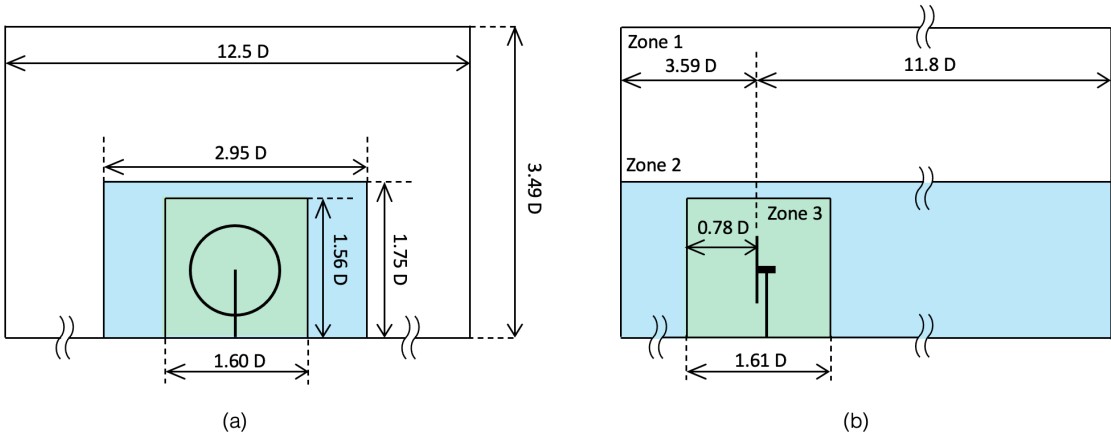

**Figure 4.** Frontal (**a**) and lateral (**b**) views of the computational domain and refinement zones used for the wind turbine simulations. Precursor simulations were used to generate turbulent inlet conditions at a plane 3.59 D upstream of the rotor disk. The cell size in the three zones is 0.055 D, 0.027 D, and 0.014 D, respectively.

For the full-scale machine, each inflow was scaled in space and time, as previously explained, resulting in flows with the same identical characteristics at the two scales. Similarly, the same LES and ALM grids were geometrically upscaled and used for the full-scale simulations; this means that also the full-scale simulations have the same slight anisotropic blockage effects of the wind tunnel case.

Figure 5 shows the streamwise velocity $\bar{u}/u_0$, where $\bar{(\cdot)}$ indicates a time-averaged quantity and $u_0$ the time-constant hub-height wind speed, at the chamber cross-section 3.59 D in front of the rotor. Panels (a) and (c) report the results of an experimental mapping of the flow performed with triple hot wire probes, while panels (b) and (d) report the numerical results; (a) and (b) correspond to the medium turbulence case, while (c) and (d) to high turbulence. Notice that measurements are available only 0.18 D above the floor. A good match between experimental measurements and simulation results can be observed over

the whole cross-section of the test chamber, including not only the vertical shear but also the slight horizontal non-uniformities. These are made even more clear by Fig. 6, which reports the Reynolds shear stress component $\overline{u'v'}/u_0^2$, where the prime here indicates a fluctuation with respect to the mean.

     For the same plane, Fig. 7a shows the mean (i.e., time-averaged) speed profile along a vertical line directly in front of the rotor center, while Fig. 7b reports the TI profile on the same line. Here again, a good match between experimental measurements

and simulations can be observed, except in the immediate proximity of the floor.

## 5    Results

### 5.1    Code to experiment verification

First, experimental measurements obtained with the G1 are compared with the corresponding numerical simulations. Two operating conditions in the partial load regime (region II) are considered: one aligned with the flow and one with a misalignment

angle $\gamma$ equal to 20 deg. Table 1 reports the experimental and simulated power and thrust coefficients in the two cases, in medium TI conditions. Notice that the power coefficient of the G1 is lower than the one of the G178. Using BEM, this difference can be fully explained by the lower efficiency of the airfoils of the scaled blade (see Fig. 2), since TSR and circulation are matched.

**Table 1.** Experimental and simulated power and thrust coefficients for the G1 turbine, in the medium TI Case.

| Coefficient | $C_P$ | | $C_T$ | |
|---|---|---|---|---|
| Case | Experiment | Simulation | Experiment | Simulation |
| $\gamma = 0$ deg | 0.416 | 0.420 | 0.881 | 0.851 |
| $\gamma = 20$ deg | 0.364 | 0.358 | 0.810 | 0.742 |

     Figure 8 shows hub-height time-average horizontal profiles of the streamwise velocity and of turbulence intensity (Wang et

al., 2019). In the experiments, wake data was measured with triple hot-wire probes at a sampling frequency of 2,000 Hz for a duration of 40 seconds, which corresponds to almost one hour at full scale. Results are reported for the aligned case at various downstream distances, for both the medium (Fig. 8a) and high (Fig. 8b) TI cases. The downstream distances are different for the two TI cases, because the data sets were obtained in previously performed unrelated experiments. While the match of the wake profile is excellent for all locations, the numerical results slightly overestimate turbulence intensity in the center of the

near wake region. Overall, simulation and experimental results are in very good agreement.

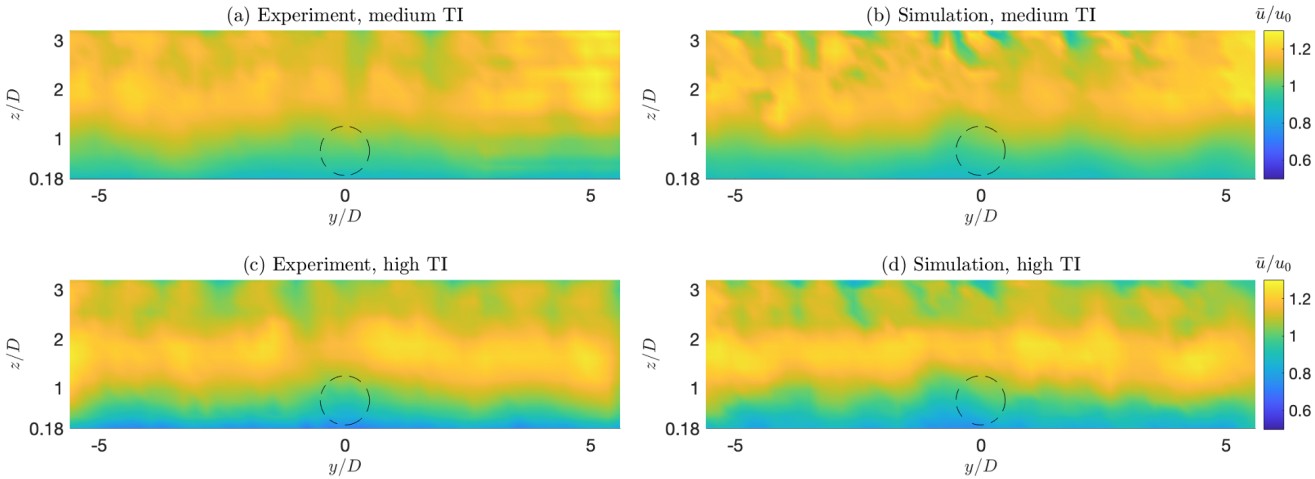

**Figure 5.** Streamwise velocity distribution on a cross section of the test chamber 3.59 D in front of the rotor. **(a, c)** Experimental measurements; **(b, d)** numerical simulations; **(a, b)** medium TI case; **(c, d)** high TI case.

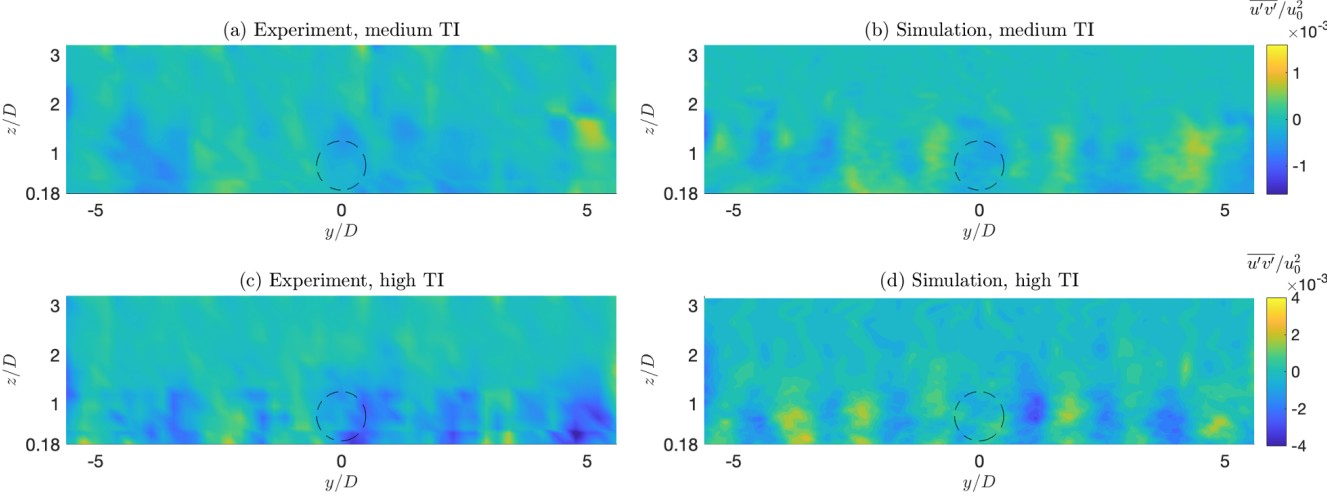

**Figure 6.** Shear stress distribution on a cross section of the test chamber 3.59 D in front of the rotor. **(a, c)** Experimental measurements; **(b, d)** numerical simulations; **(a, b)** medium TI case; **(c, d)** high TI case.

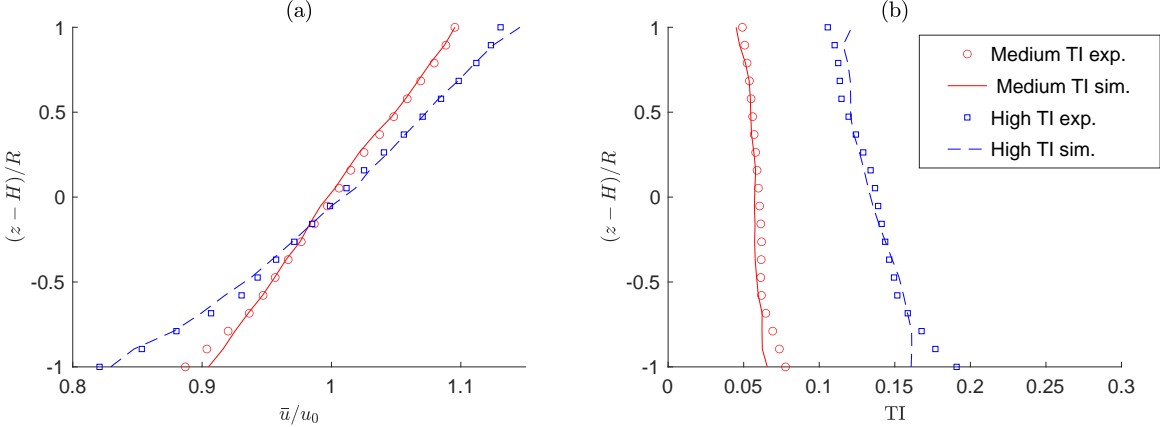

**Figure 7.** Mean velocity **(a)** and turbulence intensity **(b)** distributions along a vertical line 3.59 D in front of the rotor.

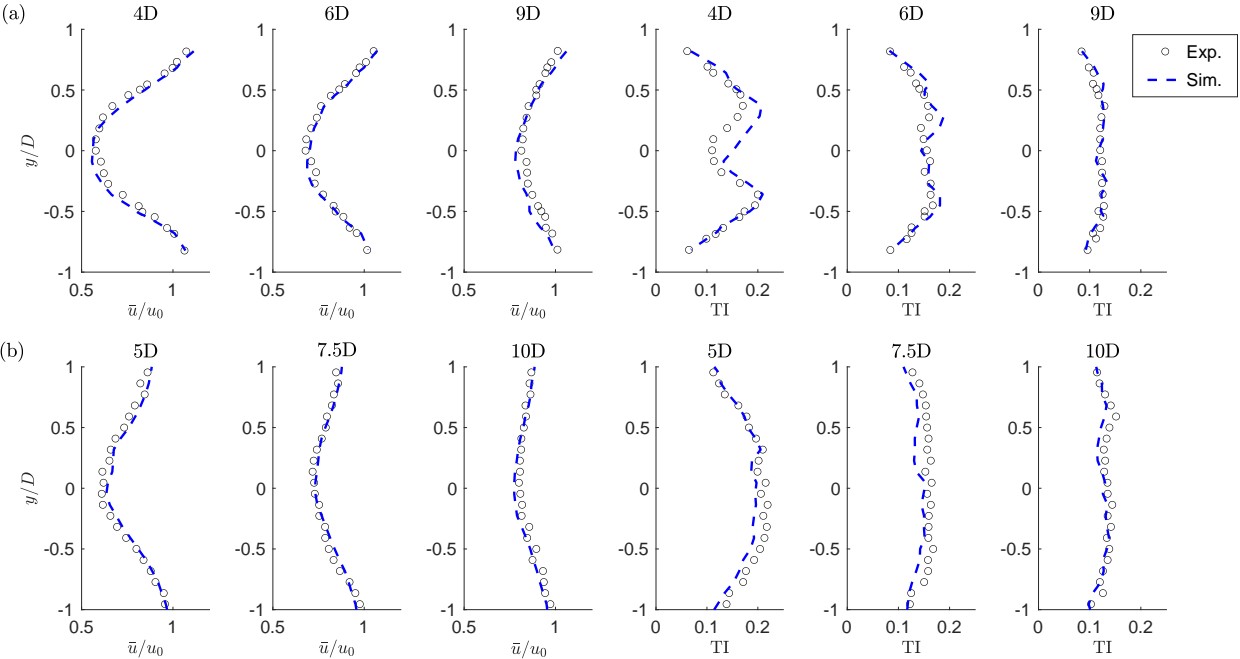

**Figure 8.** Horizontal hub-height profiles of normalized time-average streamwise velocity and turbulence intensity, for the medium **(a)** and high **(b)** inflow TI cases. Black *o* symbols: experimental results; blue dashed line: G1 simulations.

## 5.2 Scaled to full-scale comparisons

Next, having established a good correspondence between the numerical results and experimental measurements, simulations were conducted with the full-scale turbines to understand the effects of mismatched quantities.

Table 2 shows the turbine power and thrust coefficients for the different cases, considering the G1 and the three G178

turbine models. As expected, the power coefficient of the G1 turbine is lower than the one of all full-scale G178s, because of the reduced efficiency caused by the lower Reynolds number regime. On the other hand, there is a good match of the thrust coefficient, especially for G178; the nRA and MC versions produce a slightly lower lift in the inboard section of the blade, and hence have a marginally lower $C_T$.

**Table 2.** Power and thrust coefficients for the different turbine models in the two considered operating conditions.

| Coefficient | $C_P$ | | | | $C_T$ | | | |
|---|---|---|---|---|---|---|---|---|
| Turbine model | G1 | G178 | G178-nRA | G178-MC | G1 | G178 | G178-nRA | G178-MC |
| $\gamma = 0$ deg | 0.420 | 0.475 | 0.472 | 0.470 | 0.851 | 0.831 | 0.827 | 0.822 |
| $\gamma = 20$ deg | 0.358 | 0.421 | 0.418 | 0.417 | 0.742 | 0.731 | 0.727 | 0.723 |

Figure 9 gives a qualitative overview of the wakes of the G1 and G178 turbines for the aligned and misaligned cases.

The wake deficits are similar, except for the central region of the near wake, as expected. Even this qualitative view shows a significant effect of the much larger nacelle of the G1. This difference however disappears moving downstream, and the far wakes of two turbines appear to be almost identical.

A more quantitative characterization of the differences between the scaled G1 model and the realistic full-scale G178 turbine is given by Fig. 10 (medium TI) and Fig. 11 (high TI), considering the misaligned case. For both figures, panel (a) shows the

mean speed in the longitudinal direction, while panels (b) and (c) show the Reynolds stress components $\overline{u'u'}/u_0^2$ and $\overline{u'v'}/u_0^2$, respectively.

Results indicate an excellent match between the scaled and full-scale wakes, for both TI levels. Some differences only appear in the peaks of $\overline{u'u'}/u_0^2$ immediately downstream of the rotor. However, the velocity profiles are remarkably similar already at 3 D, notwithstanding the differences around the hub and the blade inboard sections between the two machines. Similar

conclusions are obtained for the aligned case.

## 5.3 Effects of unmatched inboard circulation and rotational augmentation

The effects of unmatched inboard circulation and rotational augmentation are quantified by computing the differences in $\bar{u}/u_0$, $\overline{u'u'}/u_0^2$ or $\overline{u'v'}/u_0^2$ at various downstream locations. Results are shown in Fig. 12, where differences are computed subtracting the G178 solution from the G178-MC or G178-nRA ones. As indicated by the figure, these effects are extremely small, and

possibly discernible from numerical noise only in the immediate proximity of the rotor.

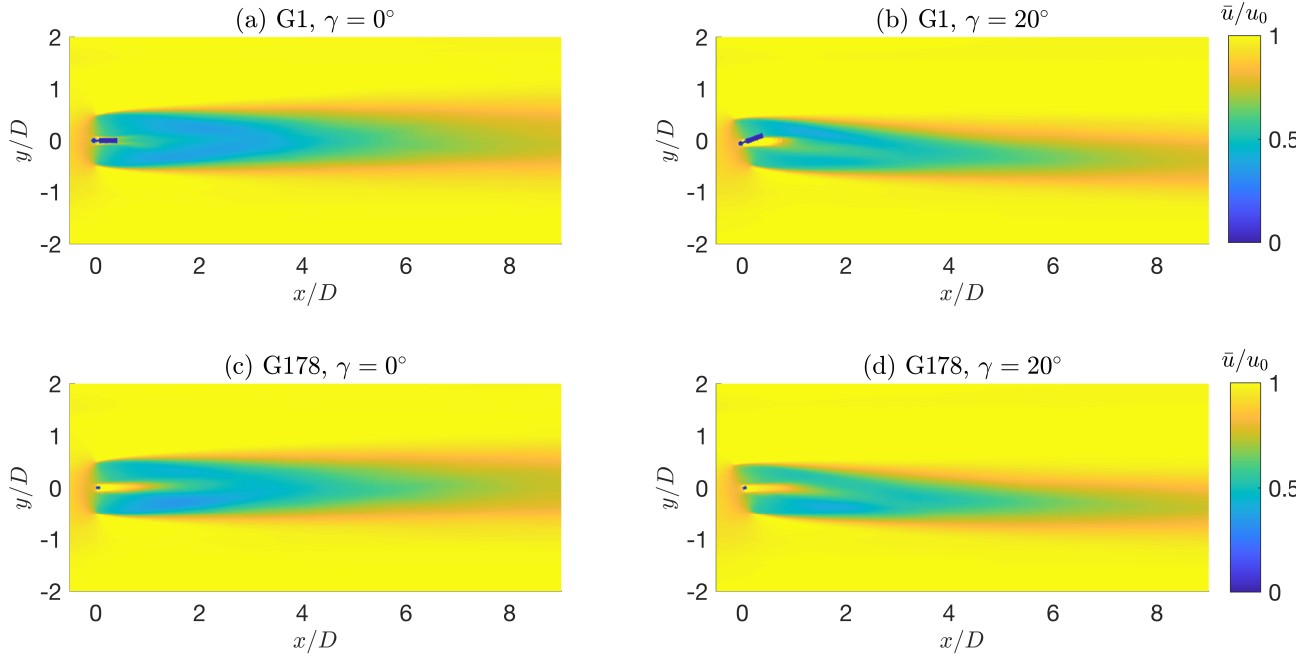

**Figure 9.** Wakes of the scaled G1 **(a, b)** and full-scale G178 **(c, d)** turbines. **(a, c)** Aligned case; **(b, d)** yaw misaligned case.

### 5.4 Effect of nacelle size and unmatched $C_P$ on swirl

For the wind-aligned operating condition, Fig. 13 shows the delta wake velocity deficit obtained by subtracting the G178-MC from the G1 solution, looking upstream. Panel (a) represents the near wake 1 D immediately behind the rotor disk plane, while panel (b) reports the far wake at 8 D. The color field represents the difference in the non-dimensional streamwise velocity deficit component $\Delta(\bar{u} - u_0)/u_0$, whereas the arrows represent differences in the in-plane velocity vectors.

In this case, since the non-dimensional circulation is matched, there are only two factors that could result in non-zero difference fields: the larger relative frontal area of the nacelle (and, similarly, of the tower) of the G1, and its smaller power coefficient caused by the chord-based Reynolds number mismatch. The impacts of these two factors are clearly visible in the near wake, respectively looking at the streamwise and in-plane velocities.

Considering first the streamwise component, the larger blockage of the G1 nacelle creates the negative velocity bubble that is clearly visible at the center of the rotor, which indicates a larger deficit behind the G1 than behind the G178-MC in this part of the wake.

The effect of the tower is different from the one of the nacelle, and leads to a positive streamwise speed difference instead of a negative one. In fact, while the nacelle is almost a pure blockage in the center of the rotor where wake recovery is the weakest,

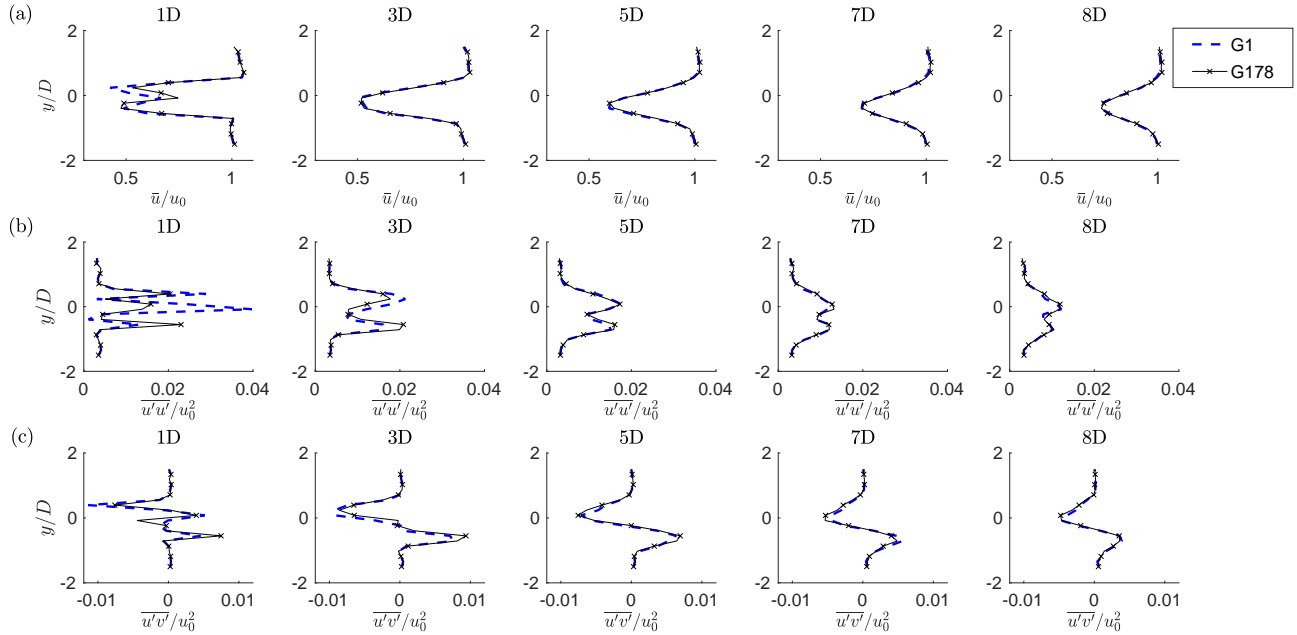

**Figure 10.** Hub-height profiles of normalized time-average streamwise velocity **(a)**, normal stress **(b)**, and shear stress **(c)**, in the misaligned and medium TI condition.

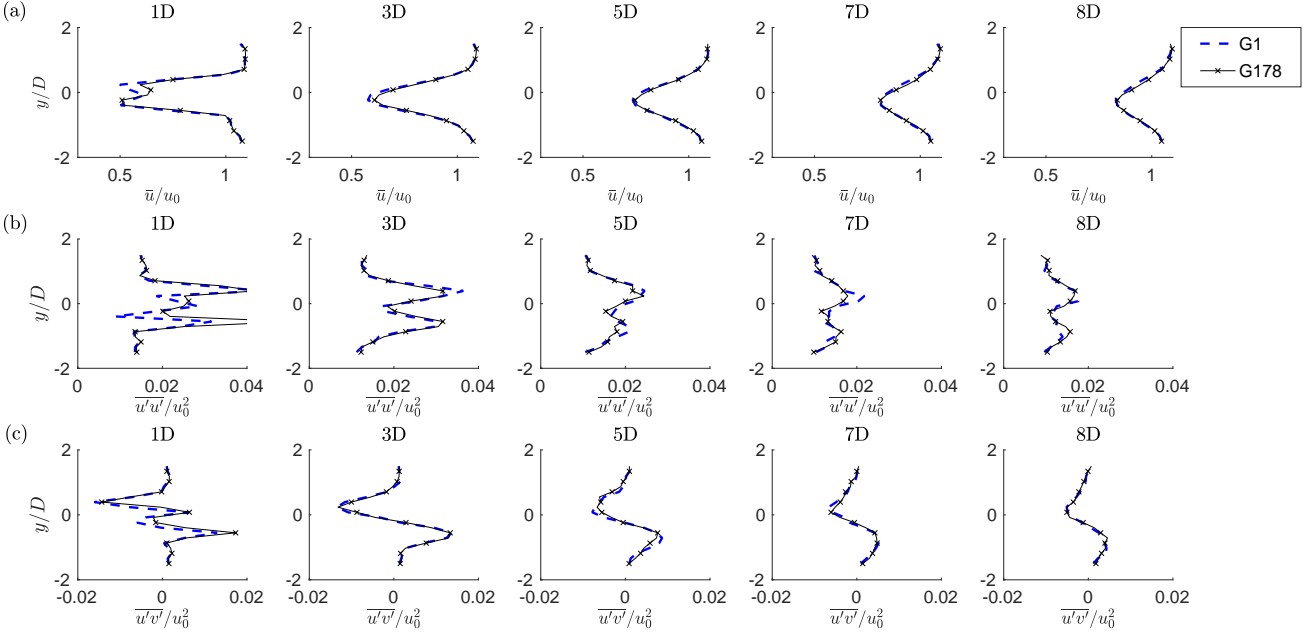

**Figure 11.** Hub-height profiles of normalized time-average streamwise velocity **(a)** and shear stresses **(b, c)**, in the misaligned and high TI condition.

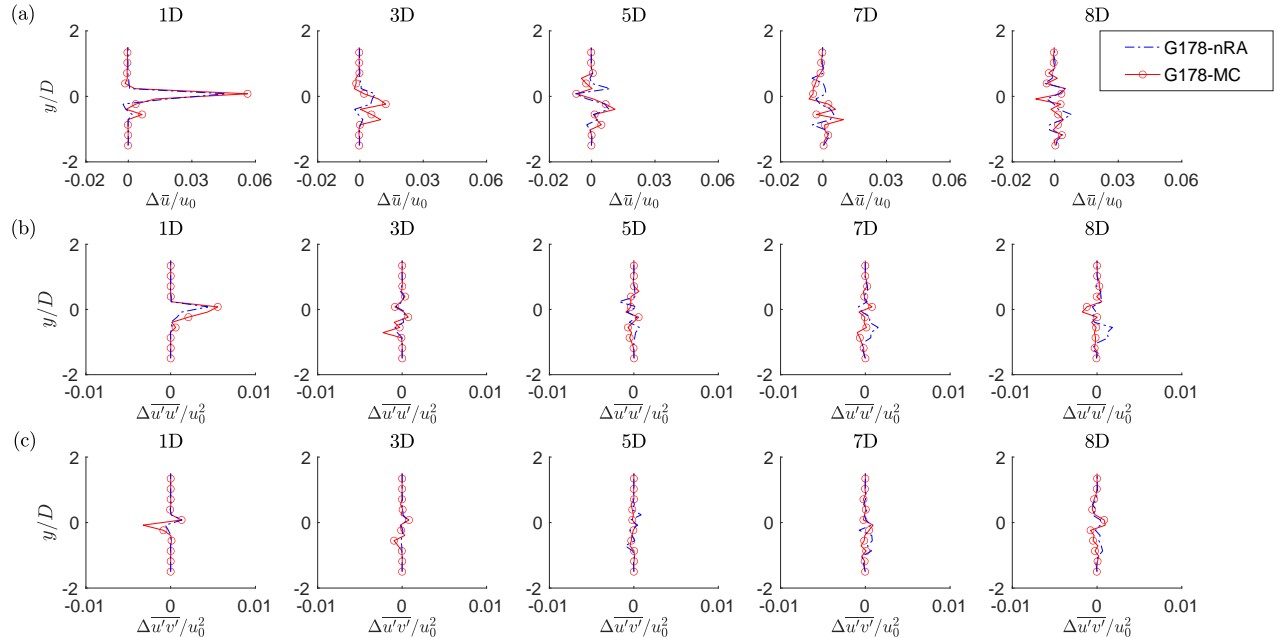

**Figure 12.** Difference in the profiles of the normalized time-average streamwise velocity **(a)**, normal stress **(b)**, and shear stress **(c)** along hub-height horizontal lines, in yaw misaligned and medium TI conditions. Dash-dotted blue line: effect of rotational augmentation, i.e. G178 results subtracted from G178-nRA results. Red solid line and ∘ symbols: effect of mismatched circulation close to the root, i.e. G178 results subtracted from G178-MC results.

the presence of the tower wake increases the local turbulence intensity, with the effect of increasing the recovery of the turbine wake. This results in the vertical region of higher streamwise speed that can be seen in the figure in the lower part of the rotor disk. When looking upstream, the rotor spins counterclockwise, whereas the wake rotates clockwise by the principle of action and reaction, and this explains why the region affected by the tower wake is convected towards the negative $y$ direction.

Consider next the in-plane velocities. Compared to the wake of the G178-MC turbine, the wake of the G1 rotates at a slower
pace, as indicated by the counterclockwise rotation of the difference field shown in the picture. The slower rotation of the G1 wake is a direct consequence of its smaller power coefficient that, for the same TSR, implies also a reduced torque coefficient. As expected, the mismatch in the swirl rotation is only concentrated close to the hub, and decays quickly with radial position.

As the flow propagates downstream and the wake progressively recovers, differences between the velocity fields decay and the effects of the mismatches can hardly be seen at 8 D. The only difference that can still be appreciated is the effect of the
larger tower. This results in some blockage close to the ground that has not yet fully recovered at this distance, resulting in about a 6% difference in the longitudinal velocity component immediately above the floor and, hence, in a slightly enhanced shear below hub height. Elsewhere, differences between the two fields never exceed 3%.

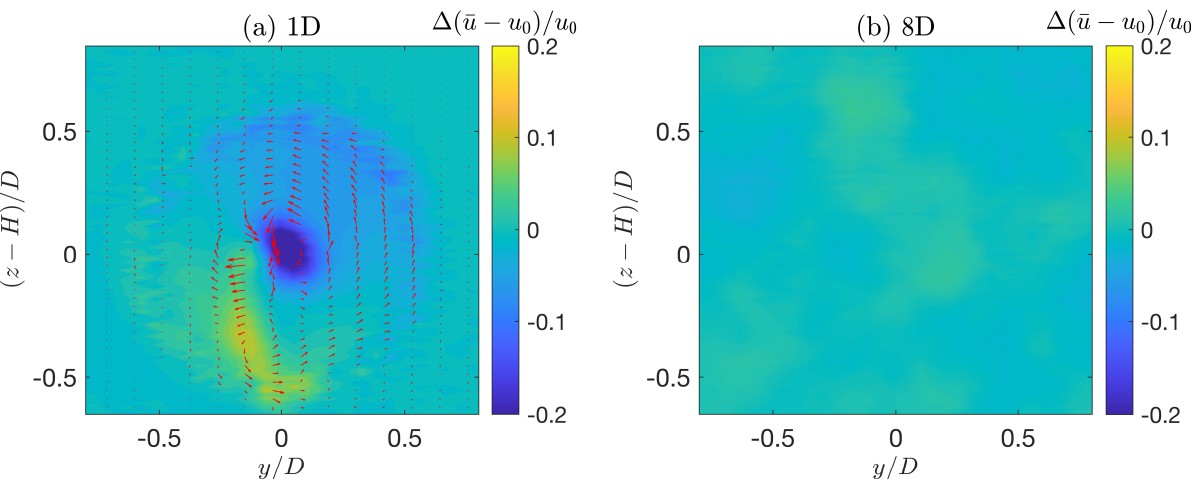

**Figure 13.** Difference in the wake velocity fields, obtained subtracting the G178-MC solution from the G1 one, looking upstream. Color field: streamwise velocity deficit difference $\Delta(\bar{u} - u_0)/u_0$; arrows: difference in the in-plane velocity vectors. **(a)** Near wake 1 D immediately behind the rotor disk plane. **(b)** Far wake at 8 D.

## 5.5 Effect of wind tunnel blockage

Considering the G1 turbine, the wind tunnel test chamber has a height $h_{wt} = 3.49$ D and a width $w_{wt} = 12.5$ D, resulting in a cross sectional area $A_{wt} = 43.6$ D$^2$. Although the resulting area ratio $A/A_{wt} = 0.018$ is relatively small, the non-negligible vertical ratio $D/h_{wt} = 0.286$ can cause some anisotropic blockage. To quantify this effect, numerical simulations were conducted in domains of increasing height from 1.75 D to 10.47 D, as shown in Fig. 14a. The actual wind tunnel height is indicated by a red square mark in the figure.

Figure 14b shows the non-dimensional power increase $\Delta P/P_\infty$ vs. the area ratio $A/A_{wt}$, where $P_\infty$ is the power for the largest domain —assumed to be blockage-free. Results indicate a power increase caused by blockage of about 1.5%.

## 5.6 Wind farm control metrics

The previous analysis has shown that the wake of the G1 turbine has a very close resemblance to the one of the full-scale G178, although some differences are present in the near wake region. However, it is difficult to appreciate the actual relevance of these differences, and a more practical quantification of the accuracy of the match would be desirable. The G1 turbine is mostly used

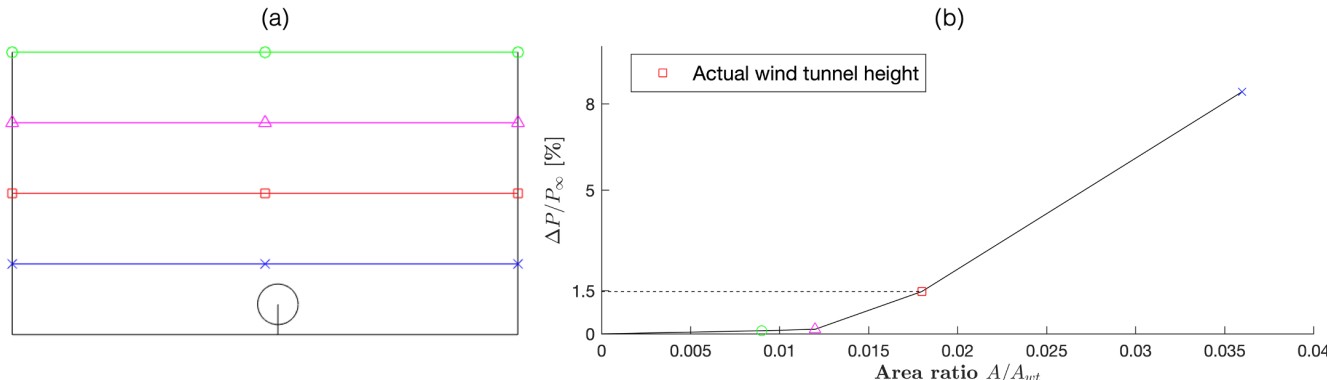

**Figure 14.** Wind tunnel blockage effect. **(a)** Cross sectional areas. **(b)** Percent power increase with respect to the unrestricted flow vs. area ratio $A/A_{wt}$.

for studying wake interactions within clusters of turbines, and for testing mitigating control strategies. This suggests the use of wind-farm-control-inspired metrics for judging the differences between the scaled and full-scale machines.

The first metric considered here is the available power ratio downstream of the turbine, noted $P_a(x/D)/P_0 = \widehat{u}^3(x/D)/u_0^3$, where $P_0$ is the power output of the turbine, and $\widehat{u}(x/D)$ is the rotor-effective wind speed at the downstream location $x/D$. The available power ratio depends on the shape of the wake, its recovery and trajectory. This quantity was computed from the longitudinal flow velocity component in the wake on the area of the rotor disk at various downstream positions directly behind the wind turbine, as shown in Fig. 15.

For the 20 deg misaligned case, the available power ratio results are reported in Fig. 16a. As shown in the figure, the available power changes moving downstream because the wake expands, recovers and —since the turbine is misaligned with respect to the wind vector— shifts progressively more to the side of the impinged (virtual) rotors. The difference of the available power behind the G1 and G178 turbines is small, and decreases quickly moving downstream. The figure also shows the effects of blockage, by reporting the results for the actual wind tunnel size using a solid line, and the ones for the unrestricted case using a dashed line; here again, this effect is very modest.

The second metric considered here is the ambient flow rotation in the immediate proximity of a deflected wake. By misaligning a wind turbine rotor with respect to the incoming flow direction, the rotor thrust force is tilted, thereby generating a cross-flow force that laterally deflects the wake. As shown with the help of numerical simulations by Fleming et al. (2018), this cross-flow force induces two counter rotating vortices that, combining with the wake swirl induced by the rotor torque, lead to a curled wake shape. As observed experimentally by Wang et al. (2018), these vortices results in additional lateral flow speed components, which are not limited to the wake itself but extend also outside of it. By this phenomenon, the flow direction within and around a deflected wake is tilted with respect to the upstream undisturbed direction. Therefore, when a turbine is

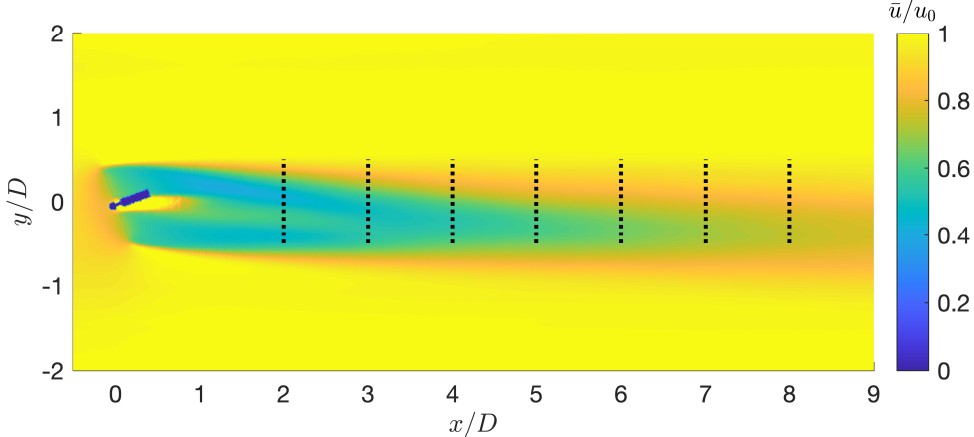

**Figure 15.** Wake of the G1 turbine for the yaw misaligned case. The black dashed lines indicate the locations of virtual downstream turbines.

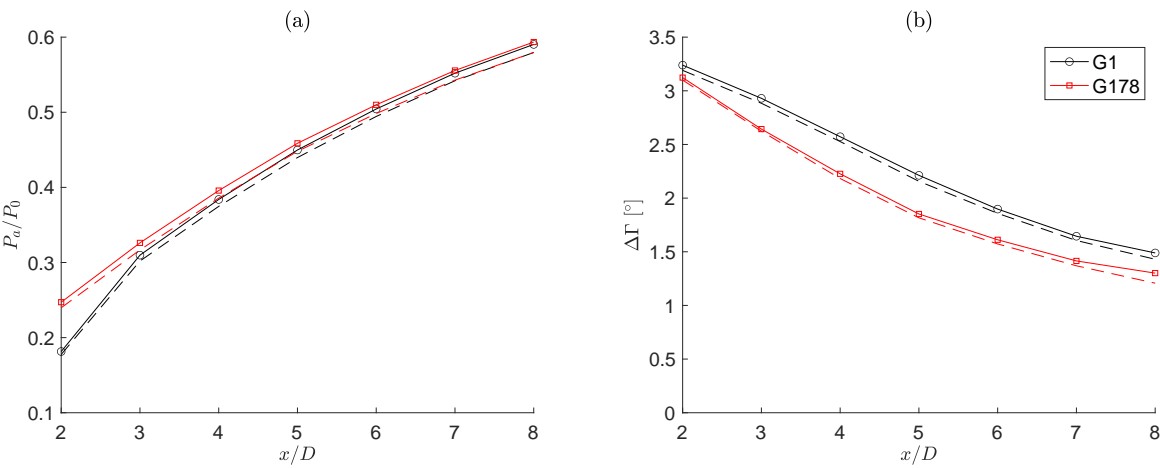

**Figure 16. (a)** Available power ratio in the wake $P_a/P_0$ as a function of downstream position $x/D$. **(b)** Change of wind direction $\Delta\Gamma$ caused by the curled wake as a function of downstream position $x/D$. Both results are for the 20 deg misaligned and medium TI case. Black $\circ$ symbols: G1; red $\square$ symbols: G178. Solid lines: actual wind tunnel size; dashed lines: unrestricted case (no blockage).

operating within or close to a deflected wake, its own wake undergoes a change of trajectory — termed secondary steering — induced by the locally modified wind direction.

The change in ambient wind direction $\Delta\Gamma$ caused by the curled wake is reported in Fig. 16b as a function of the downstream distance $x/D$; even in this case, the effects of blockage can be appreciated by comparing the solid and dashed lines. The angle $\Delta\Gamma$ was computed from the wake velocity components, averaging over the rotor disk areas already used for the analysis of the

available power. Here again the difference in the change of ambient wind direction behind the G1 and G178 turbines is quite small. A non-perfect match is probably due to the slightly different strength of the central vortex generated in response to the rotor torque. On the other hand, the two counter-rotating vortices caused by the tilted thrust are well matched —given the good correspondence of this force component between the two models.

## 5.7 Effect of integral length scale

The ILS of the wind tunnel flow was obtained by first computing the time-autocorrelation of the wind speed at one position in front of the turbine, and then multiplying the result by the mean wind speed. The length scales obtained from measurements in the wind tunnel and the simulated flow resulted in nearly identical values, as already shown by Wang et al. (2019). A second estimate of the ILS was based on the space-autocorrelation between simultaneous values of the simulated wind speed at two points in front of the turbine. For the size of the G1 turbine, this second estimate of the ILS resulted in a full-scale value

of approximatively 142 m. On the other hand, the IEC 61400-1 international standards prescribe space-autocorrelation-based lengths of 170 m in Ed. 2 (IEC 61400-1, 1999) and of 340 m in Ed. 3 (IEC 61400-1, 2005). Although the ILS presents a significant natural variability at each location and across different sites (Kelly, 2018), the value achieved in the wind tunnel with the G1 is undoubtedly in the low range of naturally occurring scales.

To understand the effects of the partially mismatched ILS on wake behavior, two turbulent inflows were generated, differing

only in this parameter. However, the passive development through spires and bricks of two inflows with different ILS values, but exactly the same TI and vertical shear, is clearly an extremely difficult task. To avoid this complication, the turbulent flow field generator TurbSim (Jonkman, 2009) was used, selecting the Kaimal model and prescribing directly the turbulence scale parameter (see Eq. (23) in Jonkman (2009)). The resulting turbulent wind time histories were specified as Dirichlet inflow conditions for the subsequent LES-ALM simulations.

The two resulting developed CFD flows are characterized by an ILS of 176 m and 335 m, and have a vertical shear exponent equal to 0.18, a hub-height speed of 11.3 $\mathrm{ms}^{-1}$, and a TI of 6.0%. These two different flows were used for conducting dynamic simulations with the G178 turbine in a 20 deg yaw misaligned condition. Figure 17 shows the spectra of the turbulence kinetic energy components, where the Kaimal Ed. 2 result is reported in Fig. 17a, while the one of the upscaled wind tunnel flow in Fig. 17b. Whereas the streamwise components are very similar, it appears that the upscaled wind tunnel flow is slightly more

isotropic than the Kaimal Ed. 2 one.

The ILS indicates the dimension of the largest coherent eddies in the flow. Hence, the main effect of a larger ILS is that of inducing a more pronounced meandering of the wake. To quantify this effect, the instantaneous wake center was computed according to the deficit-weighted center of mass method (España et al., 2011). The standard deviation of the horizontal wake

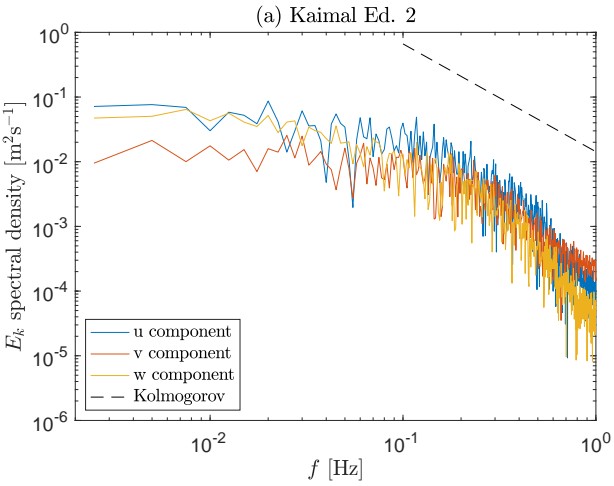 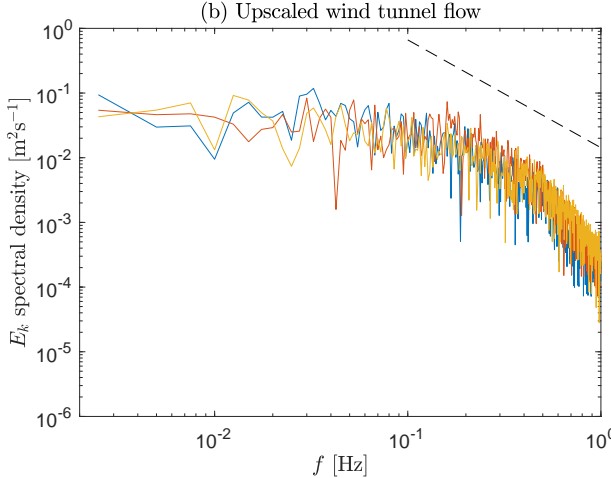

**Figure 17.** Spectra of turbulent kinetic energy components.

position 5 D downstream of the rotor was found to be equal to 0.089 D for the low ILS (176 m) case, and equal to 0.12 D for
the high ILS (335 m) one, according to expectations.

The effects of a different ILS are much smaller, although still appreciable, when considering mean quantities. Figure 18
reports the profiles of speed and shear stresses at different downstream distances. The mean velocity profile is only very
slightly affected, with a maximum change of about only 2%. A clearer effect is noticeable in the shear stresses at the periphery
of the wake.

## 6   Discussion and conclusions

This paper has analyzed the realism of wind-tunnel-generated wakes with respect to the full-scale case. In the absence of
comparable scaled and full-scale experimental measurements, a hybrid experimental-simulation approach was used here for
this purpose. A LES-ALM code was first verified with respect to detailed measurements performed in a large boundary layer
wind tunnel with the TUM G1 scaled wind turbine. Next, the same code —with the same exact algorithmic settings— was
used to simulate different full-scale versions of the scaled turbine. These different full-scale models were designed to highlight
the effects of mismatched quantities between the two scales.

Clearly, this approach has some limits and therefore falls short of providing a comprehensive answer to the realism question.
In fact, the comparison is clearly blind to any physical process that is not modelled or that is not accurately resolved by the
numerical simulations. Additionally, it is assumed that a numerical model that provides good quality results with respect to
reality at the small scale is also capable of delivering accurate answers at the full scale.

Keeping in mind these limits, the following conclusions can be drawn from the present study:

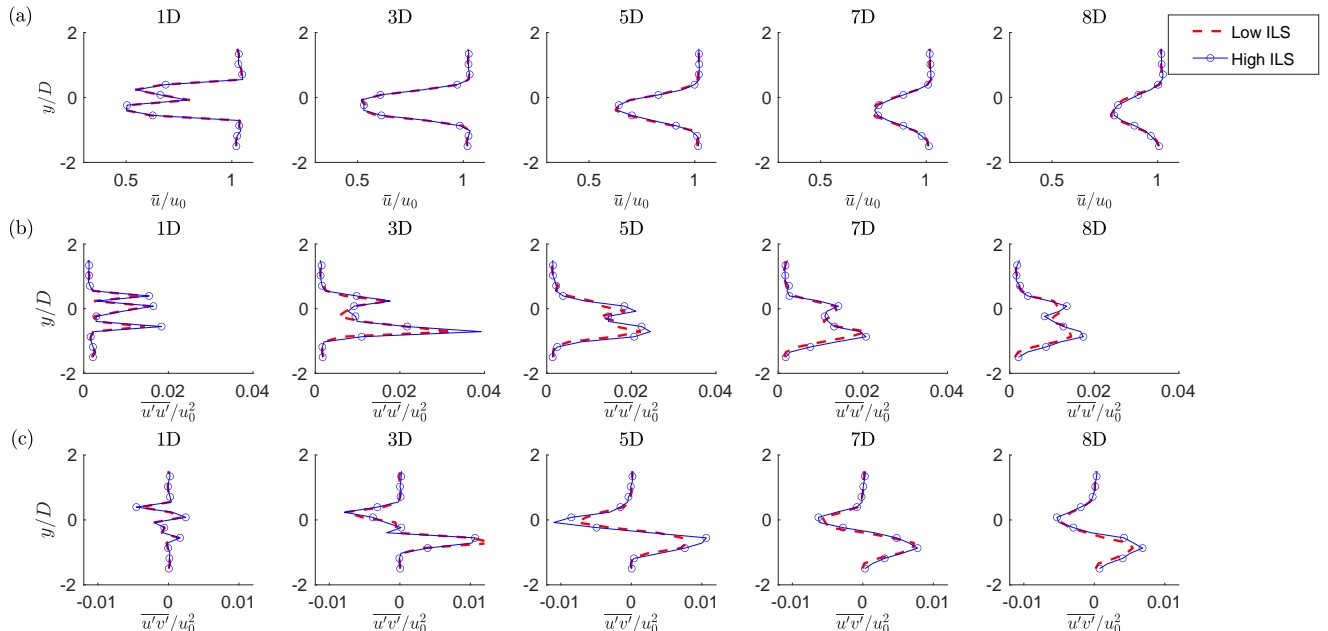

**Figure 18.** Hub-height profiles of normalized time-average streamwise velocity **(a)** and shear stresses **(b, c)**, for the low and high ILS cases in yaw misaligned conditions.

– Overall, the far (above approximatively 4 D) wake of the G1 scaled wind turbine is extremely similar to the wake of a corresponding full-scale machine considering all classical mean metrics, i.e. wake deficit, turbulence intensity, shear stresses, wake shape and path, both in aligned and misaligned conditions.

– Small differences of fractions of a degree are present in the local wind direction changes caused by the curled wake, because of a different swirl generated by the smaller aerodynamic torque of the scaled model. The trends in terms of downstream distance and yaw misalignments (not shown here) are however extremely similar.

– The effects of blockage are very limited in the large wind tunnel of the Politecnico di Milano, with differences in power of about 1.5% and negligible effects on other metrics.

– The effects of rotational augmentation, unmatched inboard non-dimensional circulation and nacelle size are clearly visible in the inner near wake region. However, they decay quickly with downstream distance, and are typically small enough not to alter the qualitative shape of the speed deficit, turbulence intensity and shear stress distributions in this region of the wake.

– The lower ILS of the flow generated in the wind tunnel at the scale of the G1 has very modest effects on mean wake metrics, although it causes a reduced meandering.

In summary, it appears that the G1 scaled turbine faithfully represents not only the far wake behavior, but also produces a very realistic near wake. This is obtained by a design of the experimental setup that matches the turbulent inflow, the geometry and strength of the helical tip vortices, and the strength and shape of the speed deficit. These are all the main physical effects dictating the evolution of the near wake. The mismatches that are present in the near-wake inner core (due to a different swirl, inboard non-dimensional circulation, rotational augmentation, and a different geometry of the nacelle) do leave a visible mark, but overall do not seem to significantly alter the behavior of the wake, as expected. The larger size of the tower leaves a more visible trace further downstream, because it affects the wake recovery by generating a local extra turbulence intensity, in turn altering shear below hub height.

Overall, the realism of both the near and far wake justify the use of the TUM G1 (and similarly designed) scaled turbines for the study of wake physics and applications in wind farm control and wake mixing.

How would these result change in case of smaller or larger scaled models? For larger models, one would still be able to match all quantities that are matched for the G1, while improving some of the unmatched quantities described in §2.3. The out-of-scale nacelle and tower of the G1 are due to miniaturization constraints of the sensors and actuators (Bottasso and Campagnolo, 2020), a problem that would be alleviated with larger models, resulting in a reduced mismatch of the vortex shedding frequency. Similarly, larger blades would reduce the mismatch of the rotation-induced stall delay and of the chord-based Reynolds number. This would lead to a better match of the power coefficient, and to improvements of some of the approximately matched quantities, such as the dynamic spanwise vortex shedding and the thrust coefficient. On the other hand, for a same wind tunnel, testing a larger model might increase blockage and the ILS mismatch. Essentially the opposite would happen for smaller models. A large chord-based Reynolds mismatch could be mitigated by increasing the rotor angular velocity, which however leads to higher power and a larger nacelle, and is eventually constrained by compressibility and by the wind tunnel speed through the TSR constraint. Additionally, one may increase solidity, although this moves the optimal TSR away from the reference (Bastankhah and Porté-Agel, 2017; Bottasso and Campagnolo, 2020). Even with very small rotors (Hassanzadeh et al., 2016), it is conceptually possible to match non-dimensional circulation and thrust coefficient, while only the latter can be matched using porous disks (Xiao et al., 2013; Lignarolo et al., 2016).

The experimental setup used in this study can be further improved, for an even increased realism and expanded capabilities. Regarding the inflow, several facilities have been recently designed or upgraded to generate unstable boundary layers (Chamorro and Porté-Agel, 2010), tornadoes and downbursts (WindEEE, 2020), or for the active generation of turbulent flows (Kröger et al., 2018). Regarding the models, a more realistic geometry and size of the nacelle and tower can be achieved at the price of a further miniaturization. Aeroelastic effects can be included by using ad hoc scaling laws (Canet et al., 2020) to design flexible model rotor blades (Bottasso et al., 2014a; Campagnolo et al., 2014). Advances in 3D printing and component miniaturization will certainly lead to advancements in the design of ever more sophisticated and instrumented models. Regarding measurement technology, a more detailed characterization of salient features of the flow can be obtained by PIV or lidars, for example in support of the study of dynamic stall, vortex and stall-induced vibrations.

Although advancements in the testing of scaled wind turbines come with significant design, manufacturing, measurement and operational challenges, wind tunnel testing remains an extremely useful source of information for scientific discovery, the

validation of numerical models, and the testing of new ideas. A quantification of the realism of such scaled models is therefore a necessary step in the acceptance of the results that they generate.

*Code and data availability.* The LES-ALM program is based on the open-source codes foam-extend-4.0 and FAST 8. The data used for the present analysis can be obtained by contacting the authors.

*Author contributions.* CW performed the simulations and analyzed the results; CLB devised the original idea of this research, performed the scaling analysis, interpreted the results and supervised the work; FC was responsible for the wind tunnel experiments and the analysis of the measurements, and co-supervised the work; HC designed the full-scale turbine models; DB validated the full-scale turbine models with BEM and CFD codes. CW and CLB wrote the manuscript. All authors provided important input to this research work through discussions, feedback and by improving the manuscript.

*Competing interests.* The authors declare that they have no conflict of interest.

*Acknowledgements.* The authors express their appreciation to the Leibniz Supercomputing Centre (LRZ) for providing access and computing time on the SuperMUC-NG System.

*Financial support.* This work has been supported by the CL-WINDCON project, which received funding from the European Union Horizon 2020 research and innovation program under grant agreement No. 727477, and by the CompactWind II project (FKZ: 0325492G), which
receives funding from the German Federal Ministry for Economic Affairs and Energy (BMWi).

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
