# Peer review of "How realistic are the wakes of scaled wind turbine models?"

_Wind Energy Science, 2020_

## Referee Comment (RC1) · Anonymous Referee #1 · 26 Nov 2020

General comments The subject dealt with by the authors is of great interest. Indeed, few studies today attempt to highlight the validity of the results obtained in wind tunnels, which are so important in the development processes of wind turbines. The paper is very well written and very well structured. The analytical approach is very good and well explained. A real effort of pedagogy is to be emphasized on the description and explanation of the different scaling. The method used for comparison (experiment – simulation) can be debated because it is not common to validate experimental results from numerical simulation. From this point of view, although the numerical simulation is detailed in previous articles, it seems to me that its description should be substantiated here. We can also note a welcome frankness on the limitations of certain hypotheses or results obtained. It seems to me that after the answer to some questions raised

below, it is quite likely to be published.

Questions on context and methodology - The authors seek to demonstrate good replicability on a reduced scale of a wind turbine wake. The results presented tend to show good reproducibility of the main wake parameters. Nevertheless, a discussion around the minimum size of the model would have been interesting. According to the authors, it is possible to validate the different parameters of similarity regardless of the size of the model? What parameters are most likely to be sizeable?

- The authors present their results for a horizontal axis wind turbine (may be specified in the title of the article ?). Do they think that the method used and the similarity parameters are the same for a vertical axis wind turbine? Are there any other problems during wind tunnel tests for a VAWT?

Scientific comments - As the authors quite rightly point out, the Reynolds numbers on the blade are very different between the reduced scale and the full scale. However, it seems to me that the paper does not provide enough detail on this point. The authors say that the profiles work on very different regimes (line 155-165). This may be true, but the ranges of Re involved must be given. Indeed, one can obtain an independence of aerodynamic coefficients above a certain number of Re (which depends on the airfoil). In this case, the flow regimes and topology (slope, Czmax) are not necessarily different?

- According to the authors, the difference between regimes comes only from the fact that the boundary layer is turbulent or not? If so, why not use carborundum (or others) to trig the boundary layer on the blade? - Concerning the Re, the authors attribute the difference of Power Coefficient to the different values of the Re. I think it would be interesting to ensure this point that the authors show the corresponding polar (numerical and experimental).

- A strong assumption is that the validation of small-scale numerical simulation provides confidence in the results obtained at scale 1. It seems to me that this assertion needs

to be substantiated. Indeed, the numerical method is presented very quickly and I have difficulty understanding how to be sure that the flow is correctly reproduced on such a large scale by LES-type methods. Quantitative information on cell size, calculation volumes and other simulation parameters should be provided. I think that's important, because simulation is here taken as a reference

- This need for precision on numerical simulation is all the more important as this simulation is used to detail the blockage and upstream turbulence effects. It is indeed known that the generation of turbulence by numerical simulations (whatever the method used) is still a topic of research in the community today.

- Line 112: The authors propose a Strouhal number based on the vortex shedding rotor. Can they explain which physical phenomena precisely corresponds to this Strouhal number?
* * *

---

## Referee Comment (RC2) · Anonymous Referee #2 · 13 Dec 2020

General Comments:

This paper presents a numerical evaluation study on how well scaled wind turbine experiments in wind tunnels represent actual wind turbine wakes. Large Eddy Simulation (LES) based predictions for the scaled turbine are first validated against wind tunnel data. The validated code is then used to simulate various full-scale turbines that are generated using scaling laws and using DTU 10 MW wind turbine as a baseline. Same code is also used to simulate the wake of the scaled turbine. The manuscript is well written in general and it does present interesting and important results regarding the representation of wakes obtained using scaled wind turbine tests in wind tunnels. Some more explanations and clarifications can be added to the text, after which the paper would be acceptable for publication.

Specific Comments:

1) Section 2.1, Inflow: In addition to the listed turbulence characteristics (i.e. TI and ILS) it would be useful if the authors also comment on the issues regarding the scaling of the turbulence spectra and Reynolds stress tensor (normal and shear stress components).

2) Section 2.1, Rotor vortex shedding: Do the authors refer to vorticity shedding throughout the blade span or to tip vortices, when they define the vortex-shedding frequency? This part needs to be defined more clearly.

3) Throughout the text I would suggest to use "Reynolds number mismatch" instead of "Reynolds mismatch" to be more precise.

4) Section 2.3: Regarding the mismatch of the Reynolds number, please comment on how tripping the boundary layer on the scaled wind turbine blade would impact this mismatch.

5) Section 2.3, Line 166, Tower and nacelle vortex shedding: The diameter of the G1 tower obviously can not be larger than that of the full-scale turbine tower and model turbine nacelle obviously can not have a frontal area larger than that of the full-scale machine. I think the authors meant relative size of the tower diameter and nacelle frontal area. Please correct and also mention how the relative size is determined (i.e. with respect to D ?). Line 370 also needs correction in this respect.

6) Line 252: Do the authors mean "design TSR" and "Non-dimensinal circulation"? If yes please correct.

7) Figure 3: Authors compare the mean (average) velocity distributions obtained from the experiments as well as the ones obtained from LES. What was the duration of the computational simulation to obtain the average results? Also what kind of sampling rate and sampling duration was used for the triple hot-wire probes during data acquisition? What about distributions of normal and shear Reynolds stress components? Do they

also look similar?

8) Line 332: Authors indicate that the operating conditions considered were at Region II. Why not region I where TSR is more or less near design value? In Region II TSR will be different than the design value.

9) Figure 5: Why are the selected downstream locations for high inflow TI cases different than those for the medium inlet TI cases? Same question here, how long was the simulation time for LES to obtain average quantitites? How was it selected and was it sufficiently long enough?

10) Regarding Table 2, please comment and elaborate on the consistent decrease of power and thrust coefficients from G178 to G178-nRA to G178-MC, both for 0 and 20 degree yaw cases.

11) Line 390: Recommend to use the actual blockage in percentage (i.e. A/Awt, which I think is more common in wind tunnel studies) instead of Awt/A. Also in Figure 11.

12) Line 441: The sentence starting with "Unfortunately ..." is not clear. I would recommend to rephrase.

13) I think the conclusions section is too long. It should be re-written, should be more concise focusing on the major conclusions of this study.

Technical Corrections:

1) Line 146: change "objected" to "object"

2) Line 151: change "smaller that" to "smaller than"

3) Line 344: Recommend to replace "different" with "lower"

4) Line 351: Recommend to use "quantitative" instead of "precise"

5) Recommend to present the axis as y/D and z/ D in Figure 3.

6) Regarding Figures 7 and 8, the second row actually is a normal stress, not a shear

stress. So please correct in the figure captions as well as in the text. Recommend to use "Reynolds normal stress" and "Reynolds shear stress".

7) Figure 9: Please enlarge axis titles, axis legends, plot title, etc. These are too small.

8) Line 366: Recommend to use "contour plot" instead of "panel".

9) Line 393: Chamorro and Porte-Agel, 2010 is not a recent reference as indicated in the text. Please correct.

---

## Author Comment (AC1) · 7 Apr 2021

**REVISION TO MANUSCRIPT DRAFT**

**Wind Energy Science Discussion**

**Title: How realistic are the wakes of scaled wind turbine models?**

The authors would like to thank the two reviewers for their time and for the useful feedback. All their inputs have been taken into consideration, and have contributed to the improvement of the paper. In addition, we have taken the opportunity of this revision to make several editorial changes in order to improve readability, and we have expanded the text at various points throughout the manuscript to improve clarity.

A revised version of the paper is attached to the present reply, with the main changes highlighted in red (deletions) and blue (additions).

A list of point-by-point replies to the reviewers' comments is reported in the following.

The Authors

**Reviewer #1**

**Numbered comments**

1. *[Reviewer] The method used for comparison (experiment – simulation) can be debated because it is not common to validate experimental results from numerical simulation. From this point of view, although the numerical simulation is detailed in previous articles, it seems to me that its description should be substantiated here.*
   **[Authors]** We have expanded the description of the CFD method in several places throughout the text.

2. *[Reviewer] On a discussion around the minimum size of the model would have been interesting. According to the authors, it is possible to validate the different parameters of similarity regardless of the size of the model? What parameters are most likely to be sizeable?*
   **[Authors]** We expanded the conclusions (now named "Discussion and conclusions") with a qualitative discussion on the impact of model size. However, a more precise investigation on the minimum model size is considered to be outside of the scope of the present paper. In particular, one should also consider other constraints and design requirements: is the model equipped with a rotor, or is a porous disk sufficient? Does the model need to be sensorized or not? Is it actuated with real-time controls? All these details might lead to different answers regarding the minimum model size, and therefore to differences in the matched quantities.

3. *[Reviewer] The authors present their results for a horizontal axis wind turbine (may be specified in the title of the article?). Do they think that the method used and the similarity parameters are the same for a vertical axis wind turbine? Are there any other problems during wind tunnel tests for a VAWT?*
   **[Authors]** We expect that similar conclusions could be obtained for any lift-based turbine, but not for drag-based machines. However, since we have not specifically investigated lift-based vertical axis turbines, we prefer not to comment on this point, which we consider to be out of scope for the present investigation.

4. **[Reviewer]** *The authors say that the profiles work on very different regimes (line 155-165). This may be true, but the ranges of Re involved must be given. Indeed, one can obtain an independence of aerodynamic coefficients above a certain number of Re (which depends on the airfoil). In this case, the flow regimes and topology (slope, Czmax) are not necessarily different?*
   **[Authors]** We expanded Section 3 with a discussion on the Reynolds number of the G1 and full-scale reference.

5. **[Reviewer]** *According to the authors, the difference between regimes comes only from the fact that the boundary layer is turbulent or not? If so, why not use carborundum (or others) to trig the boundary layer on the blade? - Concerning the Re, the authors attribute the difference of Power Coefficient to the different values of the Re. I think it would be interesting to ensure this point that the authors show the corresponding polar (numerical and experimental).*
   **[Authors]** We expanded Section 3 of the paper, which now includes the comparison between the efficiency of the reference wind turbine and G1 airfoils. A new figure now clearly illustrates the lower lift-to-drag ratios of the G1 airfoils, thus explaining the lower power coefficient of this turbine. As suggested by the reviewer, we have also added a comment on tripping (which is however ineffective for the low-camber airfoil of the scaled model).

6. **[Reviewer]** *A strong assumption is that the validation of small-scale numerical simulation provides confidence in the results obtained at scale 1. It seems to me that this assertion needs to be substantiated. Indeed, the numerical method is presented very quickly and I have difficulty understanding how to be sure that the flow is correctly reproduced on such a large scale by LES-type methods. Quantitative information on cell size, calculation volumes and other simulation parameters should be provided. I think that's important, because simulation is here taken as a reference.*
   **[Authors]** We have extended the descriptions of both the simulation code, its algorithmic parameters and the mesh discretization.

7. **[Reviewer]** *This need for precision on numerical simulation is all the more important as this simulation is used to detail the blockage and upstream turbulence effects. It is indeed known that the generation of turbulence by numerical simulations (whatever the method used) is still a topic of research in the community today.*
   **[Authors]** In the wind tunnel used in this study, turbulence is generated passively using spires and brick elements. Although we agree with the reviewer, results indicate that our approach, which is based on faithfully reproducing this experimental setup, results in the very good match shown in Figs. 5 and 6.

8. **[Reviewer]** *The authors propose a Strouhal number based on the vortex shedding rotor. Can they explain which physical phenomena precisely corresponds to this Strouhal number?*
   **[Authors]** We have rephrased this sentence, and added two relevant references.

**Reviewer #2**

**Numbered comments**

1. *[Reviewer] In addition to the listed turbulence characteristics (i.e. TI and ILS) it would be useful if the authors also comment on the issues regarding the scaling of the turbulence spectra and Reynolds stress tensor (normal and shear stress components).*
**[Authors]** We have added Fig. 16 in section 5.7 to compare the flow spectra obtained by the TurbSim code and upscaled wind tunnel simulated flow. Regarding shear stresses, these are included in figs. 9, 10, 11 and 17 for the characterization of the wakes. Regarding shear stresses at the inflow, please see the reply to question 7 below.

2. [*Reviewer*] Section 2.1, Rotor vortex shedding: Do the authors refer to vorticity shedding throughout the blade span or to tip vortices, when they define the vortex-shedding frequency? This part needs to be defined more clearly.
**[Authors]** See reply to question 8 of reviewer 1.

3. *[Reviewer] Throughout the text I would suggest to use "Reynolds number mismatch" instead of "Reynolds mismatch" to be more precise.*
**[Authors]** Thank you for the advice, we have updated the text.

4. *[Reviewer] Section 2.3: Regarding the mismatch of the Reynolds number, please comment on how tripping the boundary layer on the scaled wind turbine blade would impact this mismatch.*
**[Authors]** We have added a comment in Sect. 3, but unfortunately tripping does not provide benefits for the low-camber airfoil used on the G1.

5. *[Reviewer] Section 2.3, Line 166, Tower and nacelle vortex shedding: The diameter of the G1 tower obviously can not be larger than that of the full-scale turbine tower and model turbine nacelle obviously can not have a frontal area larger than that of the full-scale machine. I think the authors meant relative size of the tower diameter and nacelle frontal area. Please correct and also mention how the relative size is determined (i.e. with respect to D ?). Line 370 also needs correction in this respect.*
**[Authors]** Yes, the relative size is discussed, and it is determined with respect to the rotor diameter D. The phrase "larger frontal area" has been changed to "larger relative frontal area" to emphasize the relative size.

6. *[Reviewer] Line 252: Do the authors mean "design TSR" and "Non-dimensinal circulation"? If yes please correct.*
**[Authors]** Yes, this has been corrected throughout the paper.

7. *[Reviewer] Figure 3: Authors compare the mean (average) velocity distributions obtained from the experiments as well as the ones obtained from LES. What was the duration of the computational simulation to obtain the average results? Also what kind of sampling rate and sampling duration was used for the triple hot-wire probes during data acquisition? What about distributions of normal and shear Reynolds stress components? Do they also look similar?*
**[Authors]** We have added the requested details on sampling and duration. We have also added the Reynolds shear stress $\overline{u'v'}$ in the characterization of the inflow (fig. 6 in the revised manuscript).

8. **[Reviewer]** *Line 332: Authors indicate that the operating conditions considered were at Region II. Why not region I where TSR is more or less near design value? In Region II TSR will be different than the design value.*
**[Authors]** The design TSR appears in Region II, the so-called partial load region (this term is also mentioned in the text, for clarity).

9. **[Reviewer]** *Figure 5: Why are the selected downstream locations for high inflow TI cases different than those for the medium inlet TI cases? Same question here, how long was the simulation time for LES to obtain average quantitites? How was it selected and was it sufficiently long enough?*
**[Authors]** The different locations were chosen for different wind tunnel experiments performed at different times; this has been noted in the text. The duration corresponds to 360 revolutions, which ensured reaching a turbulent steady state, as it is now noted in the text.

10. **[Reviewer]** *Regarding Table 2, please comment and elaborate on the consistent decrease of power and thrust coefficients from G178 to G178-nRA to G178-MC, both for 0 and 20 degree yaw cases.*
**[Authors]** This can be explained with the efficiency of the airfoils (figure 2), and with small differences in the circulation. All this is now commented in the text.

11. **[Reviewer]** *Line 390: Recommend to use the actual blockage in percentage (i.e. A/Awt, which I think is more common in wind tunnel studies) instead of Awt/A. Also in Figure 11.*
**[Authors]** Thank you for the suggestion. The figure has been changed.

12. **[Reviewer]** *Line 441: The sentence starting with "Unfortunately ..." is not clear. I would recommend to rephrase.*
**[Authors]** This sentence has been rephrased.

13. **[Reviewer]** *I think the conclusions section is too long. It should be re-written, should be more concise focusing on the major conclusions of this study.*
**[Authors]** We believe that this is one of the most important sections of the paper, where some key take-home points are made. Eliminating this part would decrease readability: only a very careful reader, who has gone through all the material of the paper, would be able to appreciate the findings of this study. For this reason, we prefer to rename this section "Discussion and conclusions", and we have included here also the answer to question 2 of reviewer 1 on the effects of considering smaller or bigger models.

**Technical corrections:**

1) **[Reviewer]** *Line 146: change "objected" to "object"*
   **[Authors]:** Implemented
2) **[Reviewer]** *Line 151: change "smaller that" to "smaller than"*
   **[Authors]:** Implemented
3) **[Reviewer]** *Line 344: Recommend to replace "different" with "lower"*
   **[Authors]:** Implemented
4) **[Reviewer]** *Line 351: Recommend to use "quantitative" instead of "precise"*
   **[Authors]:** Implemented
5) **[Reviewer]** *Recommend to present the axis as y/D and z/ D in Figure 3.*

*[Authors]:* Implemented

6) *[Reviewer] Regarding Figures 7 and 8, the second row actually is a normal stress, not a shear stress. So please correct in the figure captions as well as in the text. Recommend to use "Reynolds normal stress" and "Reynolds shear stress".*
*[Authors]:* Implemented

7) *[Reviewer] Figure 9: Please enlarge axis titles, axis legends, plot title, etc. These are too small.*
*[Authors]:* Implemented

8) *[Reviewer] Line 366: Recommend to use "contour plot" instead of "panel".*
*[Authors]:* Implemented

9) *[Reviewer] Line 393: Chamorro and Porte-Agel, 2010 is not a recent reference as indicated in the text. Please correct.*
*[Authors]:* The word "recently" has been removed.

Thanks again for the useful feedback
The Authors

[revised manuscript text omitted]